# Diverse and asymmetric patterns of single-neuron projectome in regulating interhemispheric connectivity

Yao Fei[1,2,6], Qihang Wu[2,3,6], Shijie Zhao[1,4] ✉, Kun Song[2,3], Junwei Han[1,4] ✉ & Cirong Liu[2,3,5] ✉

The corpus callosum, historically considered primarily for homotopic connections, supports many heterotopic connections, indicating complex interhemispheric connectivity. Understanding this complexity is crucial yet challenging due to diverse cell-specific wiring patterns. Here, we utilized public AAV bulk tracing and single-neuron tracing data to delineate the anatomical connection patterns of mouse brains and conducted wide-field calcium imaging to assess functional connectivity across various brain states in male mice. The single-neuron data uncovered complex and dense interconnected patterns, particularly for interhemispheric-heterotopic connections. We proposed a metric "heterogeneity" to quantify the complexity of the connection patterns. Computational modeling of these patterns suggested that the heterogeneity of upstream projections impacted downstream homotopic functional connectivity. Furthermore, higher heterogeneity observed in interhemispheric-heterotopic projections would cause lower strength but higher stability in functional connectivity than their intrahemispheric counterparts. These findings were corroborated by our wide-field functional imaging data, underscoring the important role of heterotopic-projection heterogeneity in interhemispheric communication.

The white matter pathways provide fundamental structural support for communication among distant brain regions, resulting in elaborate configurations of functional connectivity (FC) networks[1–5]. As the largest white matter tract in placental mammals, the corpus callosum connects cortical regions of both hemispheres and is essential for interhemispheric communication. The coordination of neural activity between the two hemispheres is vital for normal brain function and behavior, disruption of which has been associated with a variety of brain dysfunctions[6–10]. Therefore, unraveling the organization of interhemispheric connections and their structure-function relationship is essential for understanding normal brain function and various diseases.

Interhemispheric connections can be either homotopic or heterotopic. Homotopic connections have been extensively characterized anatomically and functionally[11,12]. Despite its well-established role in supporting homotopic connections, the corpus callosum has also been found to convey a large number of interhemispheric-heterotopic cortical connections[13–15]. These interhemispheric-heterotopic connections, coupled with intrahemispheric (ipsilateral) connections from shared upstream regions, convey different types of information and may diversely regulate interhemispheric communication. However, due to the insufficient resolution of traditional connectivity data, it has been challenging to describe the area-, layer-, and cell-specific wiring

[1]School of Automation, Northwestern Polytechnical University, Xi'an, China. [2]CAS Center for Excellence in Brain Science and Intelligence Technology, Institute of Neuroscience, Chinese Academy of Sciences, Shanghai 200031, China. [3]University of Chinese Academy of Sciences, Beijing 100049, China. [4]Research & Development, Institute of Northwestern Polytechnical University in Shenzhen, Shenzhen, China. [5]Key Laboratory of Genetic Evolution & Animal Models, Chinese Academy of Sciences, Shanghai, China. [6]These authors contributed equally: Yao Fei, Qihang Wu. ✉e-mail: shijiezhao666@gmail.com; jhan@nwpu.edu.cn; crliu@ion.ac.cn

patterns of the neocortex[16,17], which is crucial for understanding how interhemispheric communication may be regulated.

In this study, we sought to elucidate the anatomical and functional organization of interhemispheric connections, drawing on recent advances in mesoscale neuronal tracing and large-scale functional imaging. To examine the organization of these connections, we used public tracing data from cre-line transgenic mice and sparse single-neuron labeling to map the layer-specific and cell-specific patterns of interhemispheric connections[16–18]. We found a denser, more diverse, and more asymmetrical pattern of interhemispheric connections than previously recognized, highlighting the extensiveness and diversity of heterotopic projections. Subsequently, we then proposed a quantitative metric to measure the heterogeneity of these projections and conducted computational simulations to investigate how varying levels of heterogeneity might influence the FC dynamics in brain networks. The models predicted that increased projection heterogeneity could result in weaker, yet more stable, FC among connected regions. These theoretical predictions were corroborated by analyzing the strength and stability of inter-regional FC using wide-field calcium imaging data across the entire dorsal cortex of mice in different brain states. This study presents a multi-modal map of the anatomical and functional layout of interhemispheric connections, highlighting the pivotal role of heterotopic projections in supporting interhemispheric communication.

## Results

### Single-neuron projectome reveals highly heterogeneous and asymmetric connection patterns

To elucidate the anatomical organization of interhemispheric connectivity, we performed a comparative study using two unique neuronal-tracing datasets from mouse brains. The first dataset, adeno-associated virus (AAV) bulk tracing data from the Allen Mouse Brain Connectivity Atlas (Allen population), labeled projections from multiple neurons in a localized area of virus injection, offering insights into structural connectivity at a population level[16,18]. We utilized 127 injections across 43 cortical regions in wild-type mice and 787 injections in various Cre-line transgenic mice to label projections from specific cortical layers (See Supplementary Data 1, 2 for detailed strain and experimental information) (Fig. 1A, *middle*). Conversely, the ION single-neuron data, provided by our institute (Institute of Neuroscience, ION), mapped projection patterns at a single-cell level using AAV anterograde sparse neuron labeling approach[17]. This dataset charted the projections of 6357 neurons in the prefrontal cortex (PFC), among which 3762 intratelencephalic (IT) neurons were selected for further analysis.

We concentrated on three types of connections: homotopic connections, intrahemispheric-heterotopic connections (or ipsilateral connections), and interhemispheric-heterotopic connections (or contralateral connections) (Fig. 1B). Analyzing the Allen population data, we constructed matrices to delineate these connection patterns across regions at a population level (Fig. 1C and Supplementary Fig. 1B). In the wild-type mice (Fig. 1C), we identified 1263 intrahemispheric-heterotopic projections among 1806 (43 × 42) region pairs, yielding a connection density of 69.93%. For interhemispheric projections, 637 were observed, contributing to a 34.45% density, which comprised 595 heterotopic and 42 homotopic projections, except for PREI lacking a homotopic projection. Most projections (97%) displayed stronger intrahemispheric (mean strength = 0.38) than interhemispheric strengths (mean strength = 0.16), and only 9 interhemispheric-heterotopic connections lacked the corresponding intrahemispheric projections. Connectivity patterns in Emx1-Cre mice (excitatory neurons) paralleled those in wild-type mice (Supplementary Fig. 1C, E). However, connectivity densities varied by layer, revealed by layer-specific Cre mice (Supplementary Data 1; Supplementary Fig. 1A). The intrahemispheric to interhemispheric ratio was lowest in layer 5

(2.04:1), followed by layer 4 (2.24:1) and layer 2/3 (3.3:1). Given that most layer 6 neurons are corticothalamic (CT), this layer exhibited the least cortico-cortical connections (32.12% intrahemispheric) and fewer interhemispheric projections, resulting in the highest ratio of 9.8:1. Overall, our analysis indicates a predominant density and strength in intrahemispheric-heterotopic projections over interhemispheric-heterotopic projections across all layers in population-level tracing data.

We then constructed a connectivity matrix of the PFC projections from the ION single-neuron dataset. For comparison, a similar matrix was derived from the Allen population data, focusing on PFC-originating projections (Fig. 1D). While the aggregated ION single-neuron data broadly mirrored the organizational principles observed in the Allen population data, the connections at the single-neuron level appeared denser. Specifically, the densities of intrahemispheric and interhemispheric connections in the single-neuron dataset were 94.59% and 77.80%, respectively, as compared to 72.51% and 38.27% in the Allen population data. This pattern of increased connectivity in the single-neuron data was consistently observed across different layers (Supplementary Fig. 1D, intrahemisphere: $p = 0.02$, interhemisphere: $p = 0.01$). Such findings suggest a significant underestimation of connection density in population-level tracing, particularly in the context of interhemispheric connections.

In addition to connectivity density, we found that projection patterns from individual regions or neurons exhibited greater diversity and asymmetry than previously expected. To categorize these patterns, we classified the projections based on their targets from a specific region or neuron. A region (or neuron) that projects symmetrically to bilateral homotopic regions is classified as forming "B" projections. In contrast, asymmetric projections are classified as either "I" (ipsilateral-specific) or "C" (contralateral-specific) (Fig. 1E). Based on this classification, a neuron (or region) projecting to bilateral hemispheres can be categorized into five types: (1) "IB" neurons projecting to asymmetric-ipsilateral (I) and symmetric-bilateral (B) targets; (2) "BC" neurons projecting to symmetric-bilateral (B) and asymmetric-contralateral (C) targets; (3) "B" neurons, only symmetrically projecting to bilateral targets (B); (4) "IC" neurons projecting to asymmetric-ipsilateral (I) and asymmetric-contralateral (C) targets; and (5) "IBC" neurons, projecting to asymmetric-ipsilateral (I), asymmetric-contralateral (C), and symmetric-bilateral (B) targets.

In the Allen population dataset, the connectivity matrix was characterized by only two regional types, "IB" and "IBC", with "IBC" regions constituting a minor proportion of 13.9% (Fig. 1F). However, analysis of single-neuron data revealed a substantially more diverse range of types. Among the 3692 intratelencephalic (IT) neurons examined, a limited 18.6% were identified as forming unilateral projections, classified either as "I" or "C" type. Conversely, bilateral projectors constituted a substantial majority at 81.4% (3004 out of 3692 neurons). The 3004 bilaterally-projecting neurons exhibited strong asymmetry, with IBC neurons making up 46.8%, IB neurons 26.4%, BC neurons 20.0%, and IC neurons 2.9%. Only a small fraction, 3.9%, were symmetrically projecting "B" types. This composition underscored an asymmetric bilateral pattern, indicating a higher diversity and complexity in single-neuron projections than observed at the population level.

These complexities were further corroborated by morphological analyses that revealed distinct axonal and dendritic characteristics among different neuronal types (Supplementary Fig. 2). Notably, "IBC" type neurons were distinguished by their longest projection lengths and the highest branch numbers, significantly differentiating them from other types ($p < 0.005$). This pattern of asymmetry and complexity was consistent across various layers and regions (Fig. 1G, H), with the "IBC" neurons emerging as the most abundant type, except for layer 1 and ACAv, where the IB subtype was abundant. Additionally, we conducted an in-depth analysis of

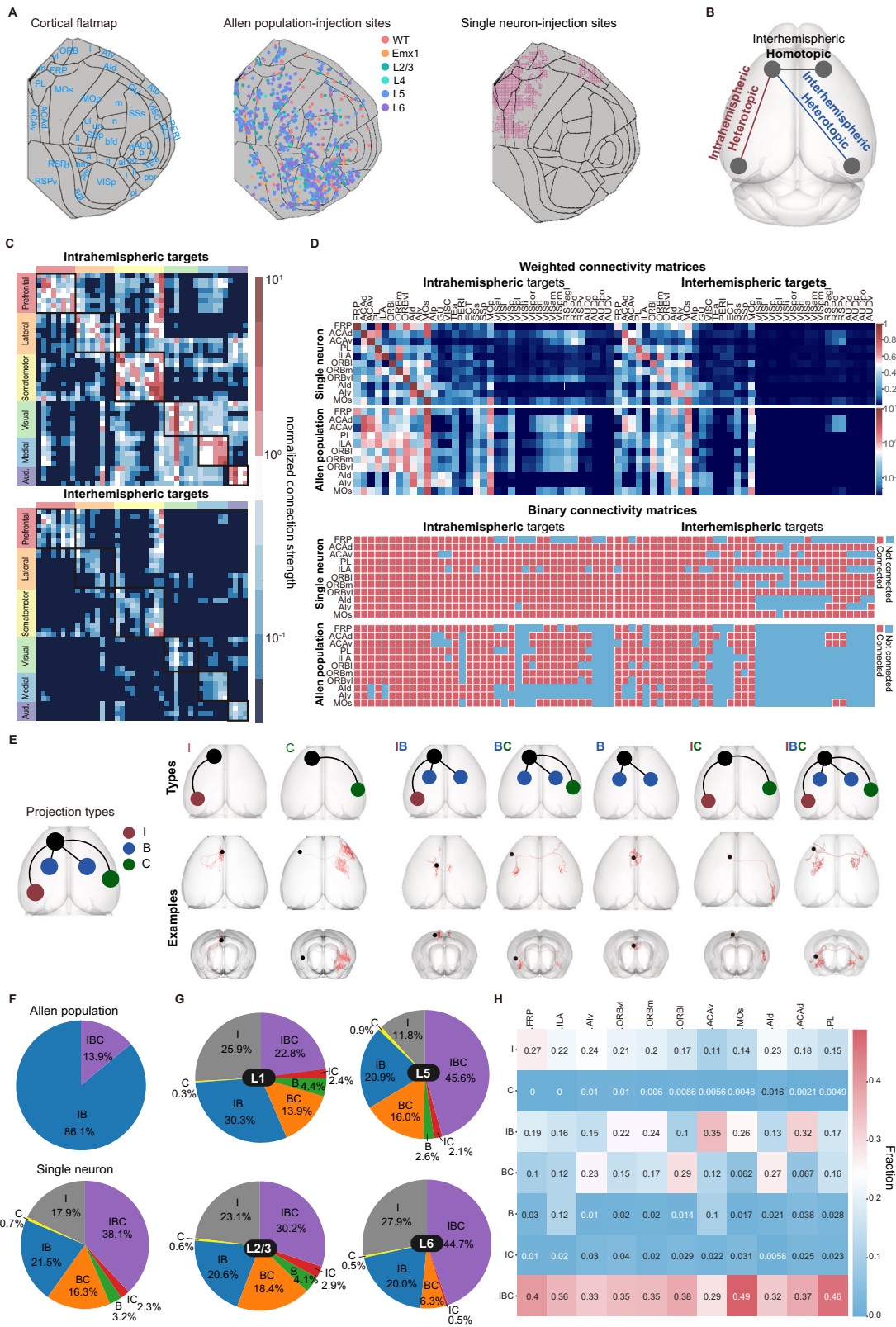

neurons exhibiting both symmetric and asymmetric projections (specifically "IB", "BC", and "IBC" types), quantifying the proportion of asymmetric projections ("I" and "C" projections) relative to total projections. Our findings revealed that asymmetric projections constituted significant proportions: 56.0% in "IB" neurons, 46.8% for "BC" neurons, and 52.9% for "IBC" neurons. This data showed that, at the single-neuron level, most neurons predominantly projected to

distinct ipsilateral or contralateral downstream targets, deviating from symmetrical projection patterns.

## Quantification of projection heterogeneity

Our single-neuron analysis revealed a dense, diverse, and predominantly asymmetric projecting pattern within the brain. Here, we introduced a quantitative metric, "heterogeneity", to measure

**Fig. 1 | Diverse patterns of interhemispheric connections. A** Two dorsal-view cortical flat maps from the CCFv3, showing 43 cortical regions and injection sites in various mouse strains from the Allen Mouse Connectivity Atlas (Allen population)[16,18]. Refer to Supplementary Data 1, and 2 for strain and experimental details, and Supplementary Data 3 for full brain region names. The third panel depicts the injection sites of ION single-neuron data[17,38]. **B** Illustrations of the connection types, including intrahemispheric-heterotopic, interhemispheric-heterotopic, and homotopic connections. **C** Connectivity matrices of wild-type mice from Allen population data, with layer-specific Cre mice data presented in

Supplementary Fig. 1C and E, F. Aud.: auditory cortex. **D** Weighted (upper two rows) and binary (lower two rows) connectivity matrices of the PFC projectome based on ION single-neuron data and Allen population data, respectively. **E** Projection types (left) and their derived seven neuron/region types. The corresponding neuron IDs of neuron-type illustration are provided in Supplementary Data 4. **F** Composition of regional types in the Allen population data (top) and neuronal types in the ION single-neuron data (bottom). **G** Laminar distribution of neuronal types from ION single-neuron data. **H** Regional compositions of different neuronal types from ION single-neuron data. Source data of (**C**, **D**) are provided as a Source Data file.

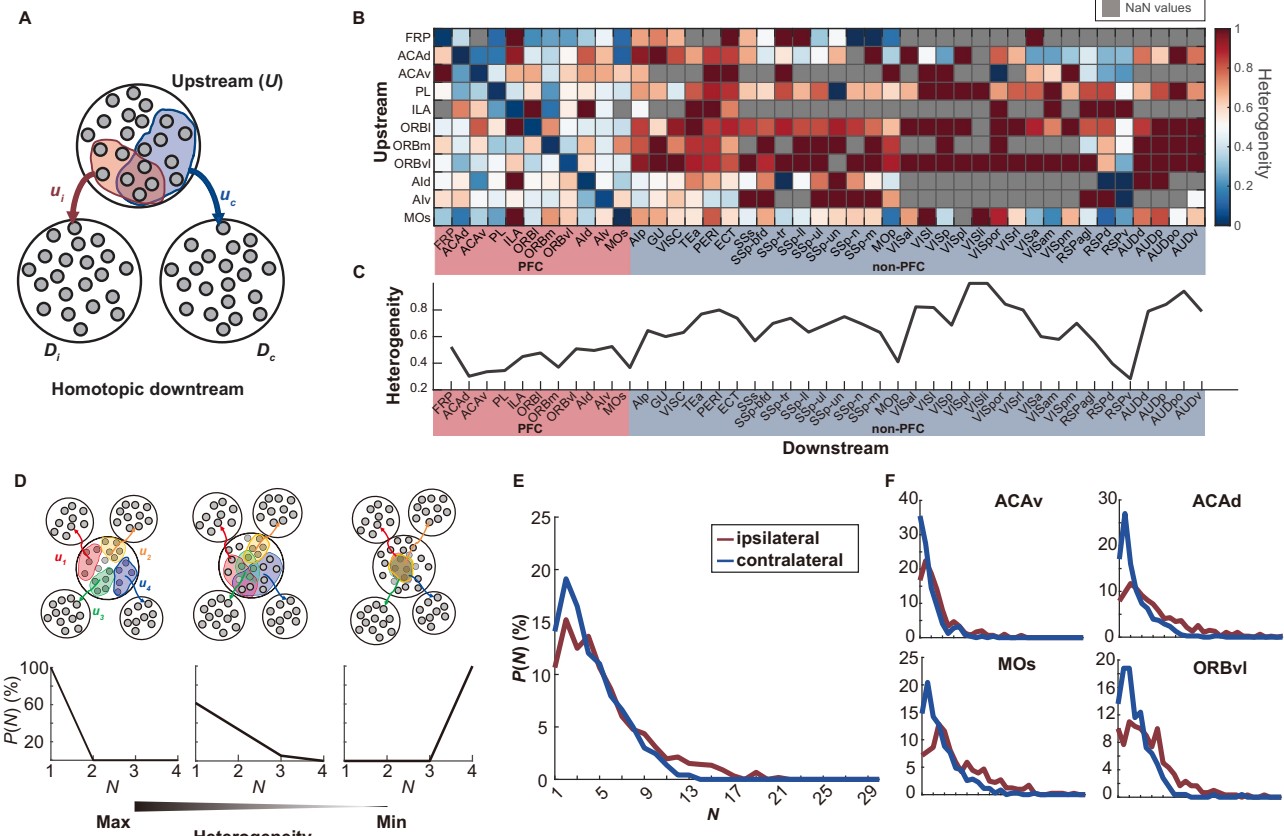

**Fig. 2 | Quantification of projection heterogeneity using single-neuron data. A** Schematic representation of the heterogeneity for the projections to a pair of homotopic downstream regions. From an upstream region $U$, one subset of neurons, $u_i$ (red), extends projections to the intrahemispheric downstream target $D_i$, while a different subset, $u_c$ (blue), projects to the contralateral downstream $D_c$. The degree of overlap between $u_i$ and $u_c$, indicated by shared neurons (magenta), is calculated as the proportion of intersecting neurons relative to the lesser of the two subsets. Projection heterogeneity is then quantified by subtracting this proportional overlap from one. **B** Projection heterogeneity among various upstream-downstream region pairings, including 11 PFC regions as upstream and 43 cortical regions as downstream, calculated from the single-neuron data. **C** Overall heterogeneity for projections to each of the 43 homotopic downstream pairs. **D** Schematic representation of the heterogeneity for the projections to multiple

downstream regions that are mutually heterotopic. The top schema delineates an upstream region projecting to four downstream regions via neuron subsets (red-$u_1$, yellow-$u_2$, green-$u_3$, and blue-$u_4$). The bottom distribution function represents $P(N = i)$, the percentage of neurons projecting to $N = i$ number of downstream regions. The values of $P$ across varying values of $N$ indicate the degree of heterogeneity: a greater proportion of neurons targeting fewer downstream regions (higher $P$ at smaller $i$) and a smaller proportion targeting more regions (lower $P$ at larger $i$) reflects larger heterogeneity. **E** Comparative heterogeneity profile of ipsilateral versus contralateral projections, calculated from single-neuron dataset across all upstream regions. **F** Heterogeneity for ipsilateral and contralateral projections from four example upstream regions: ACAv, ACAd, MOs, ORBvl. More examples are presented in Supplementary Fig. 3. Source data of (**B–E**, **F**) are provided as a Source Data file.

the patterns of the projections from a specific upstream region ($U$) to two or more downstream regions ($D$). This metric was defined by the degree of non-overlap of projecting neurons from $U$ to $D$, with higher values indicating greater projection specificity and larger heterogeneity (See *Methods, Computational modeling part, Heterogeneity of projections to pairs of homotopic regions*). We first examined the case when the downstream was a pair of homotopic regions, exemplified here by ipsilateral $D_i$ and contralateral $D_c$ (Fig. 2A). In their upstream region $U$, a subset of neurons (denoted as $u_i$) projected to region $D_i$ and another neuron subset (denoted as $u_c$) projected to $D_c$, while a subset formed projections to both

regions, as indicated by the overlap of $u_i$ and $u_c$. Heterogeneity was quantified based on the ratio of non-overlap between $u_i$ and $u_c$. Less overlap implied that neurons in $U$ had more distinct projections to $D_i$ and $D_c$, resulting in a higher heterogeneity value. We then evaluated the heterogeneity of projections from 11 PFC regions to 43 homotopic downstream pairs using the single-neuron data (Fig. 2B). The analysis revealed varying levels of heterogeneity for different upstream and downstream. Interestingly, the projections targeting downstream PFC regions showed lower heterogeneity than the projections targeting non-PFC regions, except for MOp and RSP (agl/d/v). For each homotopic downstream pair, we also combined

projections from all 11 upstream to calculate the overall heterogeneity (Fig. 2C).

In a more complex scenario where the upstream region $U$ could project to multiple non-homotopic regions, a singular metric could not capture the full complexity of the projection patterns (Fig. 2D). To address this, we assessed heterogeneity through a probability distribution function $P(N)$, representing the distribution of the number of targeted downstream regions ($N$) of each projecting neuron (See *Methods, Computational modeling part, Heterogeneity of projections to mutually heterotopic regions*). For instance, if $U$ projected to four distinct downstream regions, it comprised four neuron subsets ($u_1$, $u_2$, $u_3$, and $u_4$), each dedicated to one of the four downstream. Complete overlap among the four neuron subsets would mean that every neuron from $U$ targeted all four regions, leading to $P(N=4)$ being equal to one, signifying minimal heterogeneity (Fig. 2D). In contrast, if each neuron subset exclusively targeted a unique downstream region, $P(N=1)$ would be equal to one, representing the maximal heterogeneity. Thus, a shift of the mass of the distribution $P(N)$ towards lower values of $N$ was associated with increased heterogeneity, and vice versa. Analysis of the single-neuron data revealed that contralateral projections exhibited higher overall heterogeneity than ipsilateral projections (Fig. 2E). This difference was consistent across projections originating from various upstream regions, such as ACAv, ACAd, MOs, and ORBvl (Fig. 2F, Supplementary Fig. 3).

## Computationally modeling the impact of projection heterogeneity on neural dynamics

We then investigated how projection heterogeneity affected neural dynamics via model simulation. We employed a widely-used neuronal network model with sparse and balanced excitatory and inhibitory connections[19], which received slowly varying noisy input that mimicked the electrical and physiological perturbations on the cortical neurons[20].

We first explored how the heterogeneity of projections to a pair of homotopic regions affected the correlation between their neural dynamics. Our model consisted of multiple pairs of homotopic regions, each containing hundreds of neurons. These regions were interconnected via three types of neuronal projections: self-projections, homotopic projections, and heterotopic projections (Fig. 3A). We calculated the activity of each region by averaging the activity of all neurons within it. To tune heterogeneity in the model, we adjusted the overlap ratio of heterotopic projections as outlined in Fig. 2A (Fig. 3B, *leftmost*). For simplicity, each simulation maintained a consistent heterogeneity value across all upstream-downstream pairings. We hypothesized that heterogeneity would influence homotopic correlation, with lower heterogeneity implying increased overlap of projecting neurons and, consequently, stronger common regulation from upstream areas. Thus, we anticipated greater synchronization in the dynamics of homotopic downstream pairs as the heterogeneity of projections targeting them decreased. In addition to heterogeneity, we considered two additional structural factors that might impact homotopic correlation: (1) the strength of homotopic projections (homotopic strength), represented by the probability of neurons in a region projecting to its homotopic counterpart; (2) the overlap between neurons forming self-projections and neurons forming homotopic projections (self-homotopic overlap) (Fig. 3B, *middle & rightmost*). Changes in homotopic strength and self-homotopic overlap affected the direct structural connections between homotopic regions, which were traditionally viewed as key determinants of homotopic correlation[21–23]. Consequently, our study aimed to identify which of these three structural features exerted the most significant influence.

Through network modeling, we found that the homotopic correlations were negatively correlated with heterogeneity and positively correlated with self-homotopic overlap and homotopic strength

(Fig. 3C–E). Interestingly, the range of variation in correlation values attributed to heterogeneity exceeded those due to homotopic strength and self-homotopic overlap (Fig. 3C–E). This might be a consequence of the larger number of heterotopic projections compared to homotopic ones; specifically, in a network with $N=10$ regions (Fig. 3C–E, *upper*), each region received $(N-1) \times 2 = 18$ heterotopic projections versus a single homotopic projection. To further probe this phenomenon, we analyzed a minimal network model with just two pairs of homotopic regions ($N=2$), where the number of homotopic projections mostly approximated that of heterotopic projections. In this minimal model, the variation range caused by self-homotopic overlap and homotopic strength was comparable to that caused by heterogeneity (Fig. 3C–E, *lower*). Furthermore, the differences between the variation range caused by heterogeneity and homotopic strength or self-homotopic overlap, denoted as $\Delta$(variation range), kept expanding as the number of regions $N$ in the model increased (Fig. 3F, G). This suggested that the impact of heterogeneity became more pronounced with the increase in the number of brain regions in the network. Considering that the mouse cortex comprised numerous interconnected brain regions ($N$ was large), our results indicated that the heterogeneity of upstream projections was a dominant factor in influencing homotopic correlation, outweighing the effects of direct connections between homotopic regions.

We then examined the influence of heterogeneity on the correlation among heterotopic downstream regions. The new model variant in this part was characterized by the absence of homotopic projections, with each region being heterotopic relative to every other, thereby featuring only self-projections and heterotopic projections (Fig. 4A). In this model, we tuned heterogeneity by manipulating the degree of overlap among neuron subsets. Specifically, consider a scenario involving $N$ downstream regions, each receiving projections from a subset of $n$ neurons. If these neuron subsets had a higher degree of overlap, sharing more neurons, the total count of projecting neurons in the upstream (denoted as $m$) would be smaller. Conversely, less overlap resulted in a larger $m$. Therefore, $m$ served as an indicator of the degree of overlap among neuron subsets (Fig. 4B). To facilitate parameter tuning, we normalized $m$ to a parameter $h$, which ranged from 0 to 1 (See Fig. 4B and *Methods, Tune the structural features of the network*). This transformation allowed for the direct manipulation of heterogeneity within the network using the parameter $h$, where a higher value of $h$ corresponded to greater heterogeneity (Fig. 4C).

When projections were non-heterogeneous ($h=0$), the network alternated between asynchronous states (mean inter-regional correlation <0.95) and highly synchronous states (mean inter-regional correlation >0.95), indicating low dynamical stability. With an increase in the heterogeneity parameter $h$, there was a notable decline in the length of the synchronous states, leading to a decrease in state alternation frequency. At maximal heterogeneity ($h=1$), the network consistently maintained an asynchronous state without shifting to synchrony, indicative of increased stability (Fig. 4D, E). This pattern was observed across various thresholds for defining synchronous and asynchronous states (Supplementary Figs. 4 and 5).

Considering the asynchronous nature of the real neural activity, we evaluated the strength and variability of inter-regional correlations during asynchronous states by analyzing the mean and standard deviation of correlations over time under different levels of noise input. With increasing heterogeneity, the correlation strength decreased, indicating more non-synchronized activity (Fig. 4F). On the other hand, the correlation variability initially exhibited a slight increase at low levels of heterogeneity, but it rapidly declined with increasing heterogeneity, ultimately reaching its lowest value at maximal heterogeneity (Fig. 4G). In summary, our modeling suggested that higher heterogeneity was linked with lower strengths but enhanced stability (reduced variability) in correlations among neuronal activities of heterotopic regions.

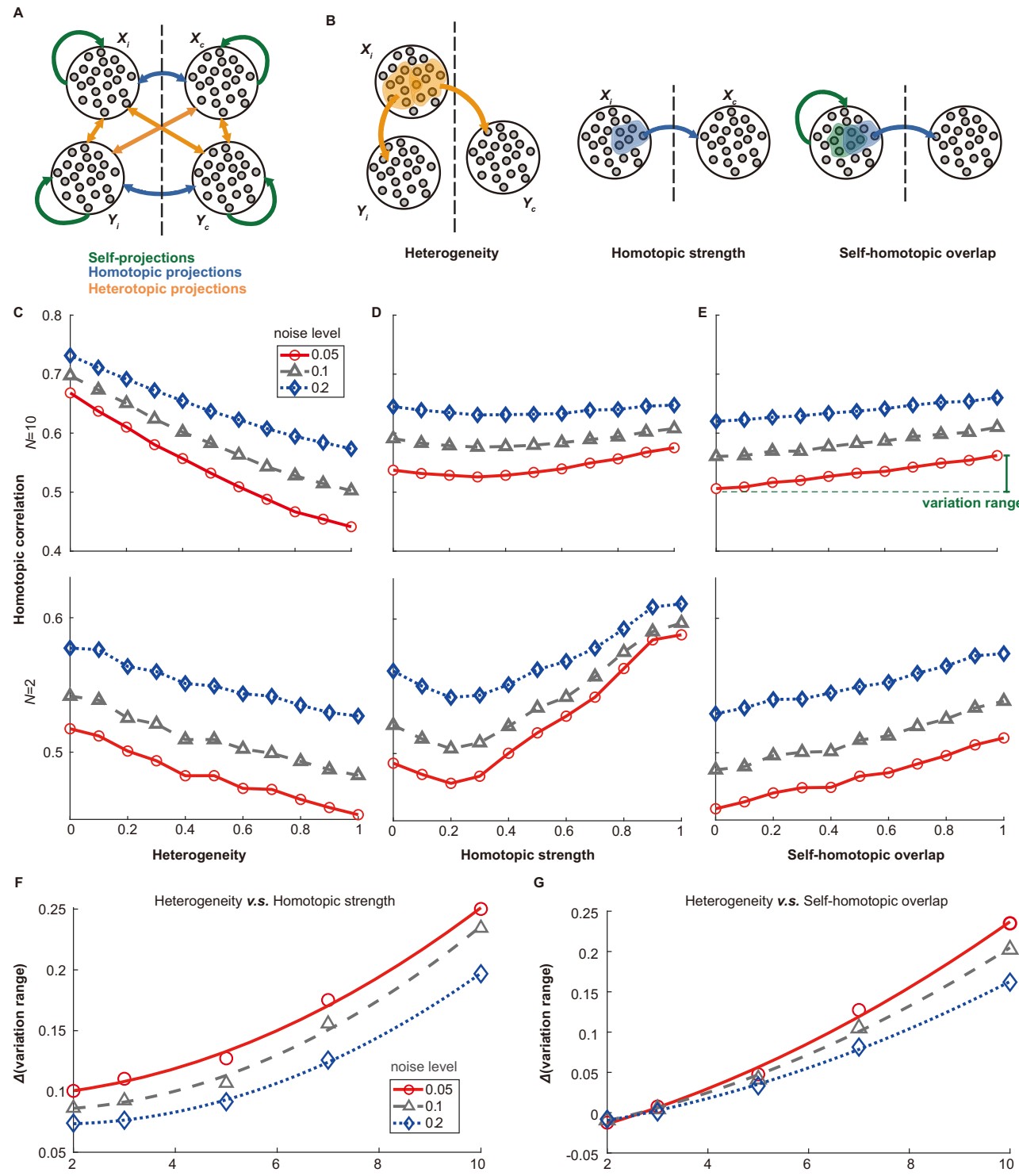

**Fig. 3 | Network dynamics modeling of homotopic region pairs. A** Diagram of the network model, including four bilateral areas ($X_i$, $X_c$, $Y_i$, and $Y_c$) and three types of projections: self-projections (green), homotopic projections (blue), and heterotopic projections (orange). **B** Illustration of the three structural features: heterogeneity, homotopic strength, and self-homotopic overlap. Homotopic correlation demonstrates a decline with higher heterogeneity (**C**), and an increase with stronger homotopic connections and greater self-homotopic overlap (**D, E**), across varying noise conditions (0.05, 0.1, 0.2). Discrepancies in the FC variation range attributable to heterogeneity against those due to homotopic strength (**F**) and self-homotopic overlap (**G**), respectively. For visualization clarity, a second-order polynomial model fits the trend lines. The difference in variation range (Δvariation range) consistently increases with larger $N$ under different noise levels (0.05, 0.1, 0.2). Source data of (**C**–**G**) are provided as a Source Data file.

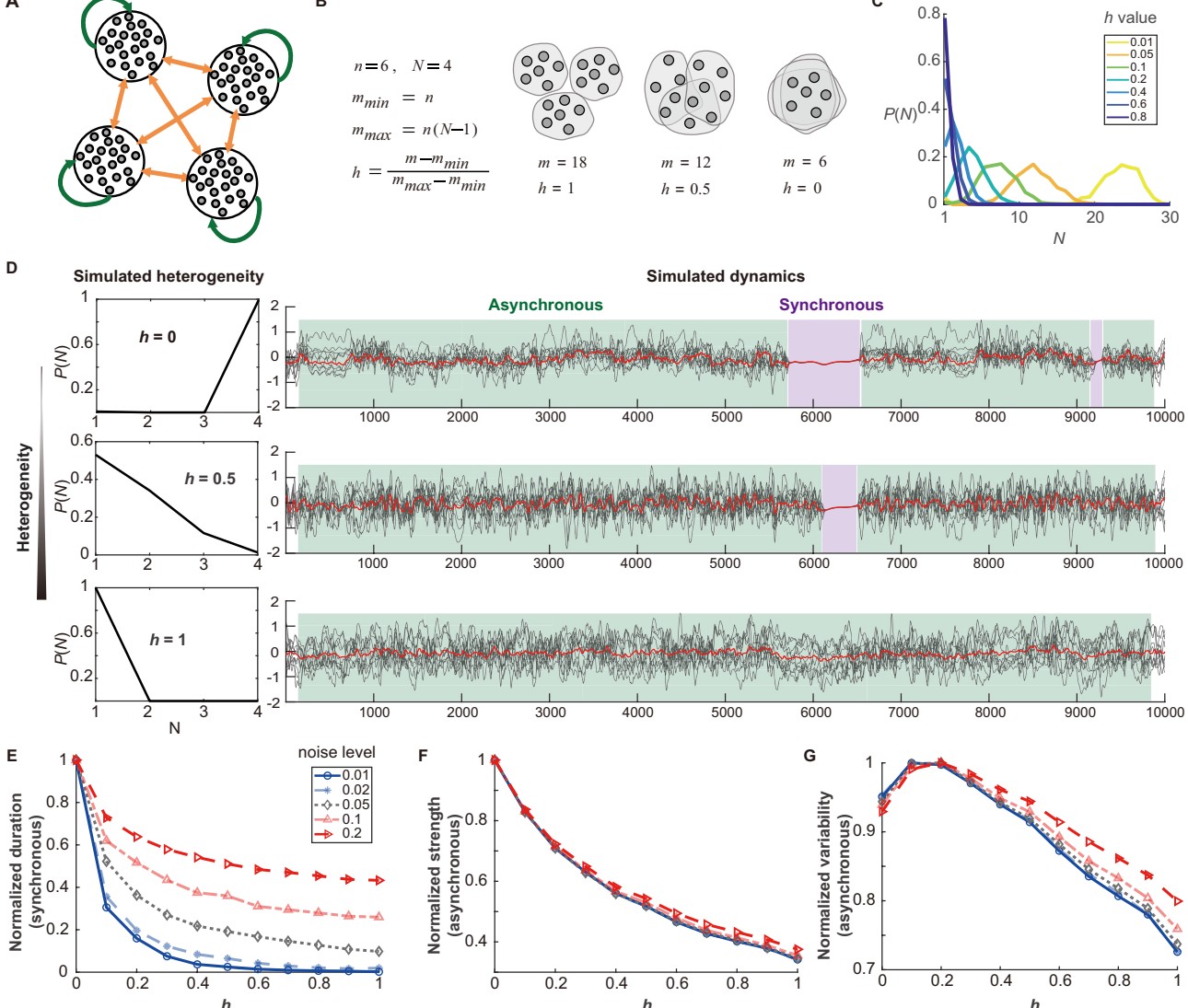

**Fig. 4 | Network dynamics modeling of multiple heterotopic downstream regions. A** Diagram of the network model with self-projections and heterotopic projections. **B** Definition of heterogeneity parameter $h$. Within an example model encompassing four regions ($N = 4$), each region contains three heterotopic downstream targets. Neuron subsets, each comprising six neurons ($N = 6$), project to respective downstream regions. The total number of projecting neurons in an upstream region ranges from its minimum value ($m_{min} = n$) when neuron subsets completely overlap, to its maximum value ($m_{max} = n(N-1)$)) when there is no overlap between subsets. Parameter $h$, normalized from $m$, scales from 0 (complete overlap) to 1 (no overlap). The diagram on the right illustrates how $h$ and $m$ regulate the overlap between neuron subsets, where each small circle represents a projecting neuron, and large gray contours delineate different neuron subsets. **C** Adjusting $h$ changes the heterogeneity level. In simulations with $N = 30$ regions, the distribution

function representing heterogeneity is shown for different $h$ values (0.01, 0.05, 0.1, 0.2, 0.4, 0.6, 0.8). An increase in $h$ shifts the distribution $P(N = i)$ leftward, indicating a rise in heterogeneity. **D** The graphs show single neuron activities (gray lines) and the mean neural activity (red lines) under a noise level of 0.05, revealing how heterogeneity influences network dynamics. In a completely non-heterogeneous scenario ($h = 0$), the network alternates between asynchronous and synchronous states. At $h = 0.5$, state alternation frequency decreases, and at $h = 1$, the network maintains at one state. **E** The duration of state transitions decreases as heterogeneity increases. Durations are normalized relative to the maximum transition duration observed at different noise levels (0.01, 0.02, 0.05, 0.1, 0.2). **F, G** Both the normalized strength and variability of inter-regional correlations during asynchronous phases exhibit a decline as heterogeneity increases. Source data of (**C**, **E–G**) are provided as a Source Data file.

## Strength and stability of interhemispheric functional connectivity

To validate our model's predictions, we examined the strength and variability of inter-regional correlations in experimental recordings of neural activity. We built a wide-field imaging system that can perform multi-channel recordings across the entire dorsal cortex of mice and synchronize the signals from electroencephalography (EEG) and electromyography (EMG) to facilitate accurate determination of mouse status (Fig. 5A). We recorded neural activities of skull-cleared Thy-1 GCamp6s mice in different natural brain states, including awake, non-REM sleep (NREM) and REM sleep. The observed brain states

exhibited different brain activity patterns (Fig. 5B), aligned with previous studies[24,25]. Although exhibiting similar patterns of inter-regional FC (Supplementary Fig. 7A), different states differed in FC strengths (Fig. 5C and Supplementary Fig. 7B). Specifically, when comparing with the NREM state, a significant increase in FC strength was observed in 91.61% of intrahemispheric connections during the awake state and in 62.80% during the REM state. In terms of interhemispheric connections, 90.22% during awake and 57.96% during REM states showed significantly greater FC strength compared to the NREM state. Furthermore, we identified that 37.42% of intrahemispheric heterotopic connections in the awake state, 42.58% in the NREM state, and 43.12%

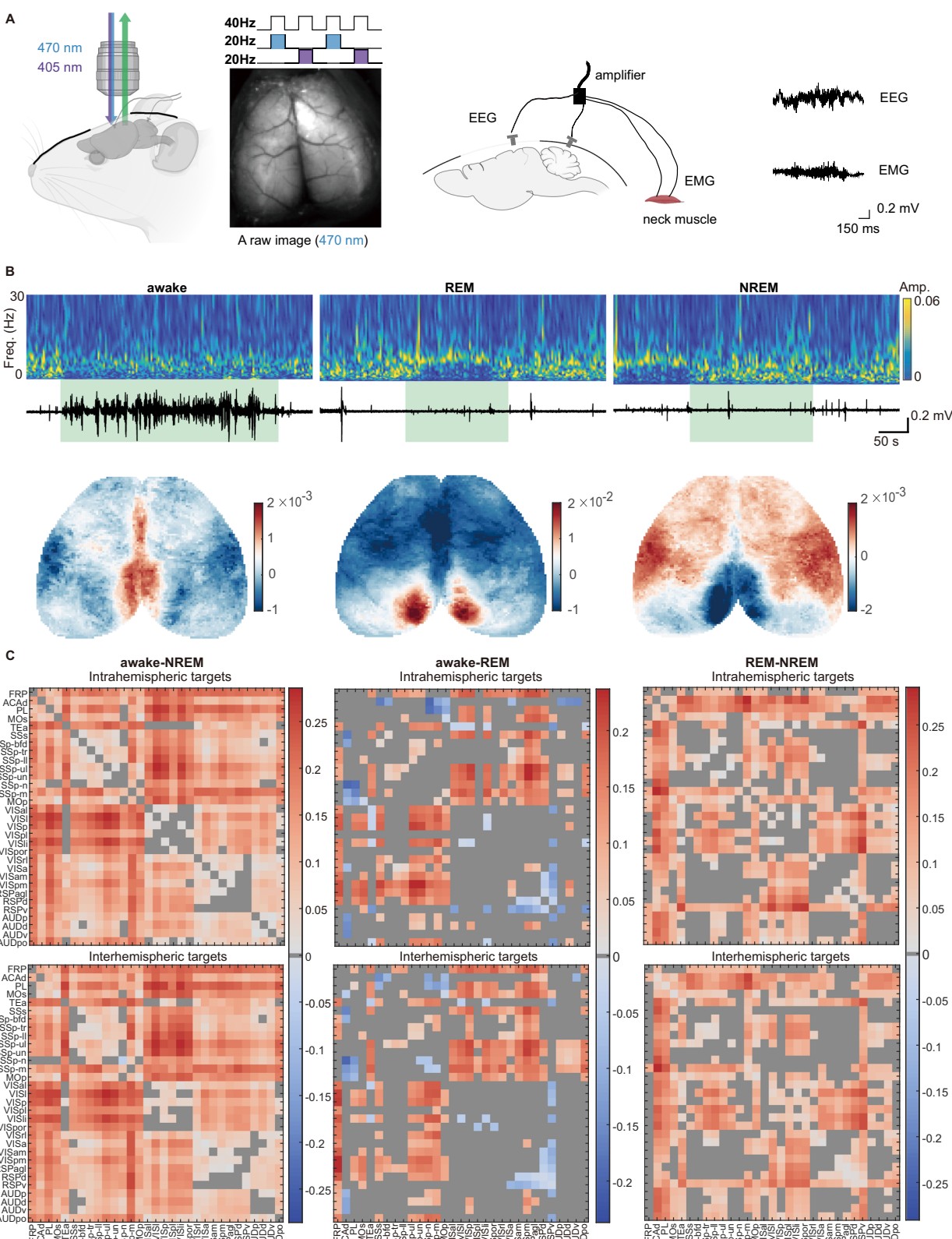

**Fig. 5 | Wide-field calcium imaging of different brain states. A** The diagram depicts wide-field calcium imaging in Thy1-GCaMP6s mice, synchronized with electroencephalography (EEG) and electromyography (EMG) recordings. To reduce hemodynamics signals (see *Methods* and Supplementary Fig. 6A), the 470 nm excitation light and 405 nm light are alternated frame-by-frame, achieving a sampling frequency of 20 Hz. Cartoons created by Biorender.com.
**B** Differentiation of brain states according to the EEG spectrum (upper) and EMG signals (bottom), with the duration of the brain state highlighted in green. Lower panels: Brain activity patterns during awake, REM, and non-REM sleep states,

represented by the mean fluorescence intensity of calcium imaging. The baseline (F) is set as the median fluorescence intensity over the entire recording time, and the color scale displays $\triangle F/F$ values. **C** Disparity matrices illustrate differences in FC strength for intra- and interhemispheric connections across various brain states. Areas shaded in gray represent connections where the FC strength differential is not statistically significant ($N = 19$, a two-sided $t$ test, $p > 0.05$). The corresponding $t$ statistic and $p$ value for all connections are provided in Supplementary Data 7. For the raw FC matrices pertaining to each brain state, refer to Supplementary Fig. 7. Source data of (**B**, **C**) are provided as a Source Data file.

in the REM state exhibited significantly greater strength than inter-hemispheric heterotopic connections (Supplementary Fig. 7A, $p < 0.05$).

The cortical FC patterns provided useful data to evaluate our model predictions. We first compared heterogeneity and the strength of direct homotopic connections in terms of their influence on homotopic FC (Fig. 6A, B). Considering that the wide-field imaging window covered 31 regions in each hemisphere, we analyzed the homotopic FC across 31 corresponding region pairs. The heterogeneity of projections to each homotopic region pair was determined using the single-neuron data (see Fig. 2C). The strength of direct homotopic connections could be derived from either the Allen population data or the single-neuron data. Since the Allen population data encompassed more brain regions compared to the single-neuron dataset, we used the homotopic connection strength estimated from the Allen population data for our analyses. Subsequently, we examined the correlations between homotopic FC and both heterogeneity and homotopic connection strength. Our findings revealed a pre-dominantly negative correlation between homotopic FC and hetero-geneity ($r = -0.614$, $p < 0.001$) (Fig. 6A), but a weak correlation between homotopic FC and the strength of homotopic connections ($r = 0.218$, $p = 0.238$) (Fig. 6B). These observations supported our model's pre-diction that heterogeneity was a key factor in influencing homotopic correlation (Fig. 3).

We subsequently evaluated the strength and variability of FC for intrahemispheric and interhemispheric connections (Fig. 6C, D). According to the model predictions, we expected the interhemispheric-heterotopic FC to be weaker but more stable since the contralateral projections were found to have larger heterogeneity (Fig. 2E, F). We measured FC among anatomically connected regions as identified in the Allen population data, employing network densities of 69.93% for intrahemispheric-heterotopic and 34.56% for interhemi-spheric connections, in line with the previously established threshold[12]. As our analysis identified a higher density of anatomical connections in single-neuron data, we also applied an adjusted net-work density threshold of 91.22% for intrahemispheric-heterotopic connections and 70.26% for interhemispheric connections (Supple-mentary Fig. 9A).

Both adjusted and unadjusted threshold analyses yielded similar results (Fig. 6C, D and Supplementary Fig. 9B, C). Homotopic con-nections exhibited the highest strength on average ($p < 0.001$), which aligned with a previous human fMRI study[11]. Interestingly, interhemispheric-heterotopic connections showed higher FC strength than intrahemispheric-heterotopic connections (Fig. 6C; $p < 0.05$). This observation might be attributed to the denser network and thus greater number of connections with lower FC strength in intrahemispheric-heterotopic connections. This assumption gained support from the finding that 87.3% (unadjusted) and 82.3% (adjusted) of interhemispheric-heterotopic connections had lower FC strengths than their corresponding intrahemispheric counterparts (Fig. 6E). Similar results were obtained when comparing FC strengths in indivi-dual brain states (awake, NREM, and REM) (Supplementary Fig. 9C–E), although in REM state the difference was not significant.

Further, we utilized the multivariate partial least squares (PLS) analysis to estimate the variability of FC across brain states, following a method applied in a prior study[11]. PLS yielded a salience value for each connection, which represented the degree of variance of this con-nection across states; thus, lower salience values corresponded to lower variability of FC between states. The PLS analysis resulted in a significant latent vector (LV) ($p < 0.001$), accounting for most cross-state covariance (75.4% for unadjusted, and 76.7% for adjusted thresholds). According to salience values in the LV, intrahemispheric-heterotopic connections exhibited the highest average variability across brain states (Fig. 6D). Interestingly, despite their relatively low FC strength (Fig. 6C), interhemispheric-heterotopic connections

demonstrated significantly lower FC variability than intrahemispheric-heterotopic connections ($p < 0.001$), no matter the connections were paired or not (Fig. 6D, F). The high stability of interhemispheric-heterotopic connections was also reflected in the comparison with homotopic connections (Fig. 6G–H). Each brain region had only one homotopic connection but multiple interhemispheric-heterotopic projections. Among 31 brain regions, only 7 regions contained at least one interhemispheric-heterotopic connection with greater FC strength than the corresponding homotopic connections, while 28 regions demonstrated interhemispheric-heterotopic connections with superior FC stability compared to their respective homotopic con-nections. These findings highlighted the previously underestimated functional stability of interhemispheric-heterotopic connections across various brain states and confirmed the model prediction that interhemispheric-heterotopic FC would be weaker but more stable than their intrahemispheric-heterotopic counterparts.

## Discussion

Utilizing single-neuron tracing data, we identified dense and highly asymmetric patterns of structural connections, especially for the interhemispheric case. In comparison with single-neuron tracing from the PFC, Allen population data underestimated network density by 22.08% for intrahemispheric connections and 39.53% for interhemi-spheric connections. This underestimation resulted from the imaging threshold necessary for AAV-based bulk neuronal tracing[16], which dif-ferentiated signal-positive pixels from the background. In our study, we adopted a threshold similar to the manually defined threshold in the original study to optimize the balance between true positives and true negatives[16]. This led to the omission of many weak and sparse connections, thereby underestimating network density. Contrastingly, single-neuron data, relying on a semi-automatic method to trace the axonal paths of individual neurons, allowed the reconstruction of these sparse but widely spread regional connections. This dispersed connection pattern in mice aligned with recent discoveries of diffuse projections in marmosets. Through high-resolution connectomic mapping of the marmoset brain, a recent study distinguished two projection patterns in the cortex and striatum: patchy and diffuse projection[26]. The patchy projection was defined by column-scale pre-cision of reciprocal and strong connectivity, whereas diffuse projec-tions presented a highly distributed pattern of relatively weak anterograde labeling. These widely spread, sparse but direct connec-tions augmented global connectivity density, potentially enhancing the efficiency and robustness of communication within the brain connectome.

The connections also exhibited greater diversity and asymmetry at the single-neuron level. Merely 3.2% of neurons projected symme-trically to bilateral homotopic regions, with the majority displaying asymmetrical patterns. From such complex patterns, we estimated the heterogeneity of projections. Network modeling suggested that this structural property laid the foundation for the upstream regulation of homotopic synchronization that was stronger than the effect of direct interaction between homotopic regions. These results may elucidate previous findings regarding the frontal cortex's resilience to unilateral impairment or inhibition[12]. In the mouse's anterior lateral motor cortex (ALM), neurons showed coordinated bilateral motor information representation. Yet, unilateral impairment or inhibition of the ALM neither disrupted neural representation in the contralateral hemi-sphere nor impaired the mouse's motor execution[12]. If the function depended solely on direct homotopic interaction, damage within one homologous area would likely disrupt its normal functions. Hence, the principal role of upstream regulation may bolster the resilience of downstream homotopic communications, emphasizing a potential upstream-downstream hierarchical relationship among brain regions. Heterogeneity also accounted for the increased stability in con-tralateral correlation compared to the ipsilateral case, serving as a

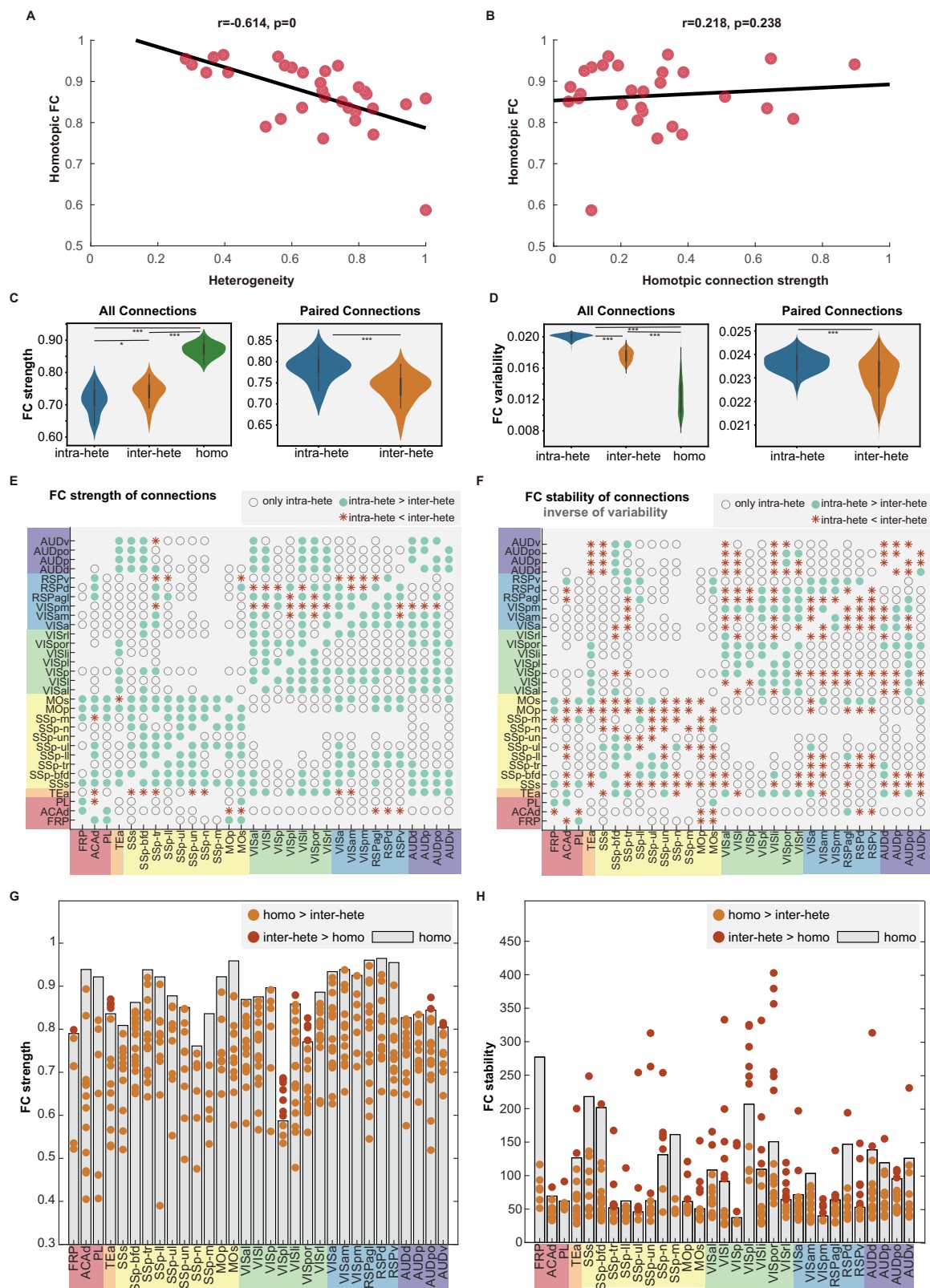

mechanism for maintaining stable interhemispheric communication of the cortex.

The robustness of interhemispheric communication was also reflected in the dynamics of FC. Typically, FC was measured by computing temporal correlations of the intrinsic infra-slow (<0.1 Hz) fluctuations of blood-oxygen-level-dependent (BOLD) signals derived from resting-state fMRI[2,11,27,28]. Through the temporal correlation of

region-wise BOLD signals, a prior fMRI study evaluated the patterns, strengths, and stabilities of FC within interhemispheric connections, revealing that homotopic connections were the most robust, with interhemispheric-heterotopic or intrahemispheric-heterotopic connections showing no significant difference in strength or stability[11]. However, recent advancements in wide-field calcium imaging have demonstrated that BOLD signals encapsulated only a small portion of

**Fig. 6 | The strength and stability of interhemispheric functional connectivity.**
**A** A significant negative correction is observed between the FC of homotopic regions and the heterogeneity of their projections ($r = -0.614$, ±95% C.I. = [−0.7955, −0.3322], $p < 0.001$). **B** Homotopic FC is not significantly correlated with the strength of direct connections between homotopic regions ($r = 0.218$, ±95% C.I. = [−0.1475, 0.5315], $p = 0.238$). **C** FC strength of the three types of connections across all brain states: intrahemispheric-heterotopic (intra-hete), interhemispheric-heterotopic (inter-hete), and homotopic connections (homo) ($N = 19$, the number of sessions). The left column presents the results of FC from all anatomically connected regions (Tukey-HSD two-side test for multiple comparisons, intra-hete vs inter-hete: $p = 0.0485$, intra-hete vs homo: $p = 0$, inter-hete vs homo: $p = 0$), which have more intrahemispheric-heterotopic connections than interhemispheric-heterotopic connections. The right column compares interhemispheric-heterotopic connections with their corresponding intrahemispheric-heterotopic counterparts (Tukey-HSD two-side test for multiple comparisons, intra-hete vs inter-hete: $p = 0.0001$). The FC strength comparison in each brain state is presented in Supplementary Fig. 7C. **D** FC variability of the three

types of connections. Variability is estimated by salience values of PLS analysis across brain states ($N = 1000$, the number of Bootstrap sampling). Left panel: Tukey-HSD two-side test for multiple comparisons, (intra-hete vs inter-hete: $p = 0$, intra-hete vs homo: $p = 0$, inter-hete vs homo: $p = 0$). Right panel: Tukey-HSD two-side test for multiple comparisons, (intra-hete vs inter-hete: $p = 0$). **E, F** Comparison between the interhemispheric-heterotopic connections and their intrahemispheric counterparts in FC strength and stability across all brain states. Results for each brain state are shown in Supplementary Fig. 9D, E. Note that stability is represented by the inverse of variability. "only intra-hete", only intrahemispheric heterotopic anatomical connections exist between two brain regions; "intra-hete > inter-hete", the intrahemispheric heterotopic connection has a higher value (strength or stability) than the interhemispheric heterotopic one, and vice versa for "inter-hete > intra-hete". Comparison between interhemispheric-heterotopic connections and the homotopic connection of each brain region in FC strength (**G**) and stability (**H**). The median value (white point), the 25th percentile, the 75th percentile, the maximum value, and the minimum value of all violin plots are provided in Supplementary Data 8. Source data are provided as a Source Data file.

functional dynamics and neuronal activities[29–33]. Based on wide-field imaging data, we found that interhemispheric-heterotopic connections exhibited stronger FC stability than their intrahemispheric-heterotopic counterparts, a difference undetectable by BOLD fMRI[11], likely due to its limited sensitivity in detecting changes in weak anatomical connections.

Our research offers a systematic delineation of the anatomical and functional patterns of interhemispheric connections, employing advanced neuronal tracing and functional imaging data. Despite uncovering noteworthy findings, our study faced several limitations that underscore avenues for future exploration. Firstly, the single-neuron tracing data was limited to the PFC of the mouse brain. Although this area presents complex connectional patterns, more comprehensive mappings of the entire mouse brain, as well as primate species[26,34–36], are indispensable for a deeper understanding of the anatomical principles of interhemispheric connectivity. Secondly, our single-neuron tracing analysis exposed diverse anatomical connectivity patterns across different cells, implying possible variations in FC within a neuronal population. While our wide-field imaging approach covered most cortical areas interconnected by the corpus callosum, it recorded brain activity at a population level without a single-neuron resolution. The emergence of enhanced optical imaging techniques and the development of genetically modified mice with calcium-signal indicators in upcoming research could enable large field-of-view recordings of single neurons and distinguishing neuronal types. Such advancements could foster a more refined comprehension of the functional organization of interhemispheric connectivity. Finally, our study described the heterogeneity in neural projections primarily in terms of overlap among projecting neurons. This approach simplified the complex interactions in neural networks, assuming random connections within brain regions and neglecting structured interactions among neuron subsets with distinct downstream projections. Such structured interactions likely existed and may significantly influence the complex relationships between different projections. Investigating these relationships requires more comprehensive single-neuron data that captures both intra- and interregional projections.

## Methods
All the experiments were approved by the Laboratory Animal Care and Use Committees from the Center for Excellence in Brain Science and Intelligence Technology (Institute of Neuroscience) of the Chinese Academy of Sciences.

### Analysis of neuronal tracing data
**Allen mouse brain connectivity data analysis.** The Allen Mouse Brain Connectivity Atlas was utilized to obtain population-level information

on axonal projections of interhemispheric connections[16,37]. Each injection experiment outlined the connectivity between the injection site and the entire brain, with the data being freely accessible at http://connectivity.brain-map.org/. For corticocortical projections, 127 cortical injections from wildtype mice and 787 injections from 26 Cre-line transgenic mice were used to map the cortical neuronal projections across different regions and layers (Supplementary Data 1 and Data 2 for detailed strain and experimental data).

Given that the neuronal tracing data consistently produced non-zero background signals, an image threshold was necessary to distinguish signal-positive pixels. In the initial study[16], a threshold of $10^{-1.5}$ of normalized projection volume was manually determined to optimize the balance between true positives and true negatives of projections for corticocortical projections. Normalized projection volume was defined as the total projection signals detected across all voxels within a structure, normalized by the total signals detected at the manually annotated injection site. Here, we adopted a similar threshold based on normalized connection strength output from the voxel-level interpolation model[37], which was defined as connection strength divided by the size of the source region. This definition mirrored that of normalized projection volume. Post-thresholding, the cumulative connection weights between any two regions were calculated to generate connectivity matrices for wild-type mice and various layer-specific Cre mice.

To understand regional projection patterns, the projection areas were initially categorized into three groups: unique ipsilateral cortical areas (projecting exclusively to this area in the same hemisphere without projections to the corresponding area in the opposite hemisphere; I for short), shared bilateral cortical areas (projecting to the same area in both hemispheres; B for short), and unique contralateral cortical areas (projecting exclusively to this area in the opposite hemisphere without projections to the corresponding area in the same hemisphere; C for short). Two distinct regional types were discerned, comprising the IB regional type that projected to specific ipsilateral and common bilateral targets; and the IBC regional type that projected to specific ipsilateral, specific contralateral, and common bilateral targets.

**ION single-neuron projectome data analysis.** Single-neuron projectome data of the mouse prefrontal cortex were provided by the Center for Excellence in Brain Science and Intelligence Technology (Institute of Neuroscience; ION), Chinese Academy of Sciences (https://mouse.braindatacenter.cn/)[17,38]. The data mapped axon projections of a total of 6357 neurons in multiple brain regions, including frontal pole (FRP), orbitofrontal (ORB), prelimbic (PL), infralimbic (ILA), anterior cingulate (ACA), dorsal and ventral agranular insular (AId/v) and secondary motor (MOs) areas. Among the 6357 neurons,

dendrites of 1920 neurons were also reconstructed. Projection data were preprocessed and registered to the Allen Mouse Common Coordinate Framework version 3 (CCFv3) template space, as described in a previous study[17]. Based on the Allen Mouse CCFv3 atlas, the specific areas corresponding to each neuronal soma and the terminal points of its projections were determined. This allowed for the computation of the connection strength between 11 prefrontal cortex areas (PFC) and 43 other cortical areas. If a source region comprised $M$ neurons, with $N$ neurons projecting to a target region, the connection strength between these regions was defined as $N/M$. For connections between different layers, only neurons within a particular layer in the source region were considered, and the connection strength of these neurons to the target regions was assessed. Note that IT neurons were used to study PFC projections to cortical regions.

Given the three categories of projection regions, we proceeded to calculate the proportion of each neuron's occupancy within these three projection types. Five types of neurons were identified, including IB neurons that projected to specific ipsilateral (I) and common bilateral (B) targets; BC neurons that projected to common bilateral and specific contralateral targets; B neurons that symmetrically projected to common bilateral targets; IC neurons that projected to specific ipsilateral and specific contralateral targets; and IBC neurons that projected to specific ipsilateral, specific contralateral, and common bilateral targets. To enhance the visual representation of soma distribution for each neuron category, their locations were mapped onto a dorsal view of a cortical flat map.

To delineate the axonal and dendritic characteristics of each neuron type, both the branch number and the lengths of axons and dendrites were computed for each neuron. A branch point was classified as a node with two or more offspring nodes, with the branch number equating to the total of these points. Each neuron possessed multiple terminals projecting onto the cortex. The length from the neuronal soma to each cortical terminal was defined as an axonal length and its path to reach each terminal was determined. For the terminal $m$, the Euclidean distances between adjacent points on the path were then computed and summed, resulting in the axonal length ($Length^m$) of the neuron's projection to that terminal. This is represented by the following formula:

$$Length^m = \sum_{i}^{n} ||x_i - x_{i-1}||_2, \qquad (1)$$

where $i$ = 2, 3,..., $n$, and $||\bullet||_2$ signifies the Euclidean distance between two points, m being the m$^{th}$ path. The maximum axonal length measured the neuron's most distant cortical projection, and the total axonal length of the neuron measured the sum of all axonal projections from the neuron. The total length of the neuron's dendritic projections was computed using a similar approach to that of the axons. Subsequent comparisons involving axonal morphologies and dendritic morphologies were subjected to statistical analysis using the Tukey Honestly Significant Difference (HSD) test, a two-sided examination, followed by adjustment through Holm's multiple comparison method. The statistical analysis results are provided in Supplementary Data 5.

We adopted the dendritic subtypes defined in a previous study that identified 38 subtypes according to morphology[38]. For clarity, we assigned codes to these 38 subtypes. Specifically, the 24 dendritic subtypes associated with typical pyramidal neurons were labeled from 1 to 24, adhering to the nomenclature in the referenced study. The seven dendritic subtypes of layer 2/3 atypical pyramidal neurons, previously classified as subtypes 1 to 7, were labeled from 25 to 31. The dendritic subtype of layer 4-like spiny stellate neurons was designated as 32. Additionally, the five dendritic subtypes of layer 6 atypical pyramidal neurons were labeled from 33 to 37, and a rare dendritic subtype was assigned the label 38.

## Computational modeling

The quantification of projection heterogeneity and all the model simulations were performed in MATLAB (2020a).

**Heterogeneity of projections to pairs of homotopic regions.** Let $U$ represent the upstream region and $D$ denote the downstream, with $D_i$ and $D_c$ denoting the bilateral homotopic areas of $D$. The subset of neurons in $U$ projecting to $D_i$ (or $D_c$) is represented by $u_i$ (or $u_c$). The heterogeneity of projections to $D_i$ and $D_c$ is defined as one minus the overlap between $u_i$ and $u_c$, where the overlap is calculated as follows:

$$overlap = \frac{|u_i \cap u_c|}{\min(|u_i|, |u_c|)}, \qquad (2)$$

where $|X|$ denotes the number of elements in a set $X$. For example, $|u_i|$ represents the number of neurons in the subset $u_i$. Heterogeneity is calculated as one minus overlap. The overlap takes the maximum value of 1, and heterogeneity takes the minimum value of 0 when one subset is completely included in the other subset. On the contrary, when $u_i$ and $u_c$ have no intersection, overlap takes a minimum value of 0 and heterogeneity is equal to 1.

**Heterogeneity of projections to mutually heterotopic regions.** Let region $X$ project to regions $Y_1, Y_2,..., Y_n$, which can be either intrahemispheric or interhemispheric to $X$. The corresponding sets of projecting neurons within $X$ were denoted as $x_1, x_2,..., x_n$. Heterogeneity was defined by the distribution $P(N)$, which described the number of subsets ($N$, ranging from 1 to $n$) that included the projecting neurons. With this definition, heterogeneity was also a measure of the distribution of the number of downstream regions for each projecting neuron. For instance, in an extreme scenario, where each neuron in $X$ is projected solely to one downstream, $P(N=1)=1$ and $P(N=i) = 0$ for $i$ from 2 to $n$. Larger heterogeneity represents the situation in which the mass of the distribution $P(N)$ moves to the side of smaller $N$.

**Simulate network dynamics.** The dynamics of the neuronal network evolved based on the interaction between neurons and the noisy external input. The activity of a given neuron $i$ was denoted by its membrane potential $V_i(t)$, which exhibited spontaneous linear relaxation towards resting potential in the absence of input from other neurons or the external input. The evolution of neuronal dynamics was described in the following equation:

$$\frac{dV_i}{dt} = -rV_i + \sum_{i} \mathbf{W}_{ij}f_j + V_{rest} + I(t), \qquad (3)$$

where $r > 0$ is the rate of linear relaxation of membrane potential, $\mathbf{W}_{ij}$ is the directed connection from neuron $j$ to neuron $i$, $f_j$ is the firing rate of neuron $j$, $V_{rest}$ is the resting potential and $I(t)$ is the noisy external input. The firing rate $f$ was calculated from the membrane potential $V$ according to a nonlinear transformation: $f_j = 0.5*[1+\text{erf}(\beta V_j)]$, where erf(.) is the Gaussian error function. The model considered both excitatory and inhibitory interactions, reflected in the sign of the elements in connection matrix $\mathbf{W}$. The relative amount of excitatory connections versus inhibitory connections was set by a parameter $0 < f < 1$. The strength of excitatory connections and inhibitory connections were randomly sampled from Gaussian distributions centered at $\mu_e$ and $\mu_i$ with standard deviation $s_e$ and $s_i$, while all the excitatory connections were then required to be > 0 and inhibitory connections < 0. Moreover, we made sure that the network was balanced by requiring $\Sigma \mathbf{W}_{ij} = 0$.

Model parameters for plots in Fig. 3C–G: $f = 0.8$, $\mu E = 0.1$, $\mu I = -0.4$, $sE = 0.02$, $sI = 0.04$, $p = 0.1$, $pself = 0.5$, $q = 0.1$, $Vrest = -0.1$, $r = -1$, $\beta = 10$.

Model parameters for plots in Fig. 4D–G: $f = 0.8$, $N = 5$, $\mu E = 0.1$, $\mu I = -0.4$, $sE = 0.02$, $sI = 0.04$, $p = 0.25$, $pself = 0.5$, $q = 0.1$, $Vrest = -0.1$, $r = -1$, $\beta = 10$.

**Tune the structural features of the network.** Heterogeneity and other structural features were reflected in the connection matrix **W**. The model of paired homotopic regions incorporated $2N$ homotopic areas from $N$ heterotopic brain regions. For example, if we set $N = 3$, resulting in a total of 6 areas, and each area encompassed $n = 100$ neurons, thus the network included $2Nn = 600$ neurons, making the size of **W** to be 600*600. Several parameters were used to decide the pattern of **W**. The parameter $q$ sets the sparsity of projections so that if a neuron projects to a region, it will project to $nq$ neurons in this downstream. The parameter $p_{self}$ sets the probability of self-projecting neurons so that $np_{self}$ neurons in each region will project to the region itself. Similarly, the parameters $p_{contra}$ and $p$ set the probability of homotopic projection and heterotopic projections respectively. The parameter $overlap_{homo}$ sets the ratio of the number of neurons that simultaneously form self-projections and homotopic projection over the minimum number of neurons forming self-projections or homotopic projections, as described in the *Methods* part *Heterogeneity of projections to pairs of homotopic regions*. Similarly, the parameter $overlap_{hetero}$ sets the proportion of neurons that simultaneously project to a pair of homotopic downstream regions that are both heterotopic to the upstream. Importantly, $overlap_{hetero}$, $overlap_{homo,}$ and $p_{contras}$ determine the three structural features investigated in the main results: heterogeneity, self-homotopic overlap, and homotopic strength. Higher heterogeneity, self-homotopic overlap, and homotopic strength are achieved by setting smaller $overlap_{hetero}$, larger $overlap_{homo}$, and larger $p_{contras}$, respectively. In each simulation of a set of given parameters, $overlap_{hetero}$ is kept the same for heterotopic projections from all upstream to all downstream pairs, and $overlap_{homo}$ and $p_{contras}$ are kept the same for all homotopic projections and self-projections.

The model of multiple heterotopic regions comprised $N$ areas, each corresponding to a different brain region. Hence, this model only included self-projections and heterotopic projections. For example, if we set $N$ equal to 3, with each area incorporating 100 neurons, the connection matrix **W** will have a size of $300 \times 300$. Heterogeneity in this model was also determined by the overlap between projecting neurons. Consider an upstream region $U$ and $(N\text{-}1)$ downstream regions $D_1, D_2, \ldots, D_{N-1}$, each receiving projections from a subset of $n$ neurons in $U$. We first randomly selected $m$ neurons that represented the entire union of projecting neurons in $U$, where $m$ was larger than $m_{min} = n$ and smaller than $m_{max} = (N\text{-}1)n$. Then, from the $m$ neurons, we randomly selected $n$ neurons to project to each downstream $D_i$, while ensuring that each of the $m$ neurons was selected at least once. We defined $h = (m - m_{min})/(m_{max} - m_{min})$. Therefore, $h = 0$ when $m = m_{min}$, representing the case of complete overlapping of projecting neurons and lowest heterogeneity; $h = 1$ when $m = m_{max}$, representing the case of complete non-overlapping and highest heterogeneity.

**Calculation of strength and variability of inter-regional correlation.** The activity of each region was calculated as the average over the activities of all neurons in the region. The correlation between activities of the two regions was calculated at each time point over a window of 100 time points. At each time point, the correlations between different pairs of regions were then averaged to give an overall correlation level at the time. According to the distribution of the overall correlation values over time, we decided on a threshold for dividing asynchronous periods and synchronous periods (Supplementary Fig. 4). Times when the overall correlation was larger than the threshold belonged to the synchronous periods, and otherwise belonged to asynchronous periods. The strength and variability of correlation under fixed structural features were calculated during the asynchronous periods. The strength of correlation was the average of the overall correlation over time, and the variability of correlation was the variance over time.

## Acquisition of wide-field calcium imaging data
**Animals.** Eight Thy1-GCaMP6s transgenic adult male mice (JAX024275) were procured from the Jackson Laboratory and then bred in the SPF class mouse house at the Center of Excellence in Brain Science and Intelligent Technology, Chinese Academy of Sciences. Mice were housed under strict conditions at 20–26 °C, with daily temperature difference being less than or equal to 4 °C, and 30–70% relative humidity (normally around 60%). We started wide-field imaging experiments on each mouse from the age of at least 12 weeks and ended at the age of about 5–6 months. The Thy1-GCaMP6s mice had the expression of the calcium indicator GCaMP6s in excitatory pyramidal neurons, primarily located in cortical Layers II/III and V, driven by the Thy1 promoter[39]. Standard housing conditions (22 °C–24 °C, light/dark rhythm of 7:00–19:00) were maintained for all mice, with no more than six per cage. Stringent measures were taken to observe animal welfare and minimize suffering during experiments.

**Construction of a wide-field imaging system.** A wide-field imaging system was constructed to directly record the neural activity of the entire dorsal cortex in different states of the mouse. The system consisted of three components: a microscope system, an LED light source, and an acquisition system. The imaging system used an Olympus MVX10 macro zoom microscope frame, equipped with an MVPLAPO 1X objective lens (Olympus) featuring a numerical aperture of 0.25 and a working distance of 65 mm. The system was illuminated by a triple-wavelength LED light source (pE-300 ultra, CoolLED) spanning the ultraviolet to the infrared spectrum, including multiple TTL interfaces and a high-performance three-pass filter (69401, Chroma) for swift and stable switching between color lasers at high speeds. The TTL signal from the camera was transmitted to an electrophysiological signal amplifier (Apollo, Jiangsu EGG Biotechnology Co., Ltd.) for alignment with the EEG/EMG data. The LED light source was synchronized with the camera using a high-frequency multi-channel trigger box (Multi-Stream Pro, CAIRN), enabling alternation of different wavelengths. The acquisition system employed a high-resolution back-illuminated research-grade CMOS camera (Prime BSI Express, Teledyne) with 95% quantum efficiency, 4.0 megapixels, 6.5 μm × 6.5 μm pixel size, and 100 Hz sampling rate. The rolling shutter mode was employed for higher frame speed and sensitivity, and the LED light source's feeding time was limited to the reading time of each line using violet and blue light. The camera's resolution (~6.5 μm) was much higher than the complete dorsal cortex imaging requirement (~50 μm), so a combined 2 × 2 pixel array was used for data acquisition with an actual resolution of ~20 μm and improved light sensitivity by adjusting the gain to reduce data volume.

**Fabrication of EEG/EMG electrodes.** The electrodes for EEG/EMG recording were fabricated by connecting a 1.27 mm 4-pin round hole pin to a matching female connector and soldering four 0.05 mm 304 stainless steel wires to each of the four pins. The wires were covered with polyimide (PI) tubes with different lengths to distinguish the channels: 1.0 cm for ground (GND), 0.5 cm for EEG, and 1.5 cm for EMG, with a diameter of 0.1 mm, to prevent mutual contact. The tubes were secured in place with super glue (1204490, Loctite) and the connections were checked with a multimeter. Finally, the solder joints and metal parts of the row of pins were encapsulated with epoxy resin AB glue.

**Surgery for wide-field imaging.** Mice were anesthetized with 5% isoflurane and secured to a stereotaxic device. Isoflurane was maintained at 1.0–1.5%, and the mice were warmed and had their

eyes protected. The skull was exposed after shaving and disinfection of the head, by dissecting the head skin and removing excess tissue, and by gently pushing the temporal occlusal muscle with a scalpel handle. The skull surface was coated with cyanoacrylate glue followed by nail polish. The EEG and GUD electrodes were placed around the cranial pegs in the auditory cortex and cerebellum, respectively, and the two EMG electrodes were inserted into the neck muscles. The cranial window was closed with light-curing adhesive and the stainless steel head post was fixed to the cerebellum. A black opaque film cap was secured to the mouse head. The mice received three consecutive days of two intraperitoneal injections each day of 5 mg/kg carprofen and 1% ceftriaxone sodium to prevent pain and infection.

**Design of a running device allowing free-limb movements for head-fixed mice.** To reduce the effect of mouse stress during data collection, a running tray was designed for mice to move their limbs freely while their heads were fixed. The bottom of the running disk was made of a 0.3 mm thick carbon fiber sheet with a diameter of 30 cm, surrounded by a 2 cm high soft opaque black PE plastic sheet and glued with T-7000 dot drill black glue. Five SP-8 304 gimbals were fixed around the bottom of the running disk and in the center of the circle with hot melt glue.

**Data acquisition of wide-field calcium imaging with EEG and EMG recording.** The mice were pre-trained to minimize stress and were affixed with M3 screws to a head post using a custom-made optical fixation bar while in an awake state. This device minimized the pressure exerted on the mice's bodies and did not impede their movement. The mice were mounted on the recording device without the use of anesthesia, which eliminates the side effects of anesthesia. The head position of the mice was fine-tuned for comfort through adjustments to the universal cross-rotating rod holder (NT01UP12, Natter Optics) and the XYZ-axis precision displacement table (Brilliance Machinery, Shenzhen). The lens angle was also adjusted through a heavy-duty manual angular stage (customized) to ensure that the entire imaging plane was in focus. Once the desired position was achieved, all adjustable elements were secured. The lens was positioned so that the LED beam covered the entire cranial window, and a custom hood was used to prevent the laser from entering the mouse's eye. GCaMP6s signals were obtained using a bandpass filter and captured by a CMOS camera. A multi-channel trigger box controlled the alternation of the LED light source between 405 nm and 470 nm wavelengths and synchronized with the camera, and laser power was adjusted after several tests to avoid overexposure. To reduce the file size, images were stored as a $2 \times 2$ merge with a resolution of $512 \times 512$ pixels and a pixel size of approximately 20 um, with an 11-bit pixel depth and an exposure time of 25 ms, resulting in a sampling frequency of 40 Hz. A 2.8 mm focal length infrared webcam (DS-IPC-T12-1, HIKVISION) was used to monitor mouse activity and an IR narrowband industrial camera was used for recording the mice's pupil and facial whisker movements throughout the experimental procedure. The luminance in the behavior chamber was maintained at approximately 40 lux during recording.

To reduce stress in head-fixed mice during recording, they were positioned with their heads facing downwards at a 15° angle, similar to a free-ranging position, to promote comfort and sleep. After surgery, mice underwent a one-week recovery period and pre-training adaptation, where they were held and touched gently twice daily for 20 min over two days. The head-fixation training was carried out for 5 days, with the fixing time gradually increasing daily and the process being consistent at the same time every day. As the number of training days increased, the mice's stress behavior decreased, and they were eventually able to calm down quickly after fixation and fall asleep spontaneously under head fixation.

The electrodes, fabricated in-house for EEG/EMG recording, were affixed to the heads of the mice and connected to the preamplifier through a homemade cable. Data were then acquired by the Apollo Neural Recording System with a sample rate of 1000 Hz and low-pass filtering below 200 Hz. A Fast Fourier Transform was applied to the EEG/EMG data and the data were downsampled to one sample point per 5 s. The states of Wake, NREM, and REM were manually classified based on EEG/EMG characteristics.

In total, 19 sessions of wide-field imaging recording from 8 mice were collected. For each session, the mice underwent a different duration of awake, NREM, and REM sleep. Detailed information for each session is provided in Supplementary Data 6.

**Analysis of wide-field calcium imaging data**
**Data preprocessing of wide-field calcium imaging.** All data processing and analysis were performed in MATLAB (2019b). The raw images underwent compression by combining $4 \times 4$ pixels into a single mean point. This process reduced the data volume and facilitated faster computational processing, yielding pixel sizes of approximately 60–70 μm. Subsequently, random noise was filtered through the application of the "imgaussfilt" function (with a 3-pixel window), resulting in improved image quality.

The data preprocessing procedure had three stages: (1) motion correction, (2) hemodynamics correction, and (3) registration to the standard brain atlas (CCFv3 atlas). To mitigate motion artifacts, a rigid body alignment was executed to align each frame of every channel to the midpoint of the entire recording session. The raw calcium imaging signals encapsulated not only the calcium activity but also the hemodynamics signals. To eliminate the hemodynamics signals, the 470 nm excitation light, and 405 nm light were alternately applied frame by frame, with the fluorescence under the 405 nm light being largely independent of the calcium concentration[40,41]. As depicted in Supplementary Fig. 6A, the hemodynamics signal (405 nm, violet) exhibited a descending trend that impacts the raw calcium imaging signal (470 nm, blue). Consequently, the 405 nm signals were utilized to regress the 470 nm signals and were subsequently removed (black). Lastly, nine landmarks (Supplementary Fig. 6B) were identified for the registration to the Allen CCFv3 standard brain atlas, employing the MATLAB function "fitgeotrans" with the "affine" method.

**Strength of functional connectivity.** In the analysis of FC, only anatomically connected regions were considered. The structural connectivity data was obtained from a structural connectivity matrix created at the population level using Allen bulk tracing data from wild-type mice. It was found that Allen bulk tracing data tends to underestimate network density. Thus, two types of thresholds were implemented: the default threshold, as previously mentioned, and an adjusted threshold. The adjusted threshold by single-neuron data was obtained through the following process (Supplementary Fig. 9A): Initially, both the intrahemispheric $\mathbf{I}_{All}$ and interhemispheric connectivity densities $\mathbf{C}_{All}$ were determined at the primary threshold. Structural connections were subsequently extracted from the PFC regions, and the resultant intrahemispheric $\mathbf{I}_{PFC}$ and interhemispheric connectivity densities $\mathbf{C}_{PFC}$ were calculated. Concurrently, the intrahemispheric $\mathbf{I}_{cell}$ and interhemispheric connectivity densities $\mathbf{C}_{cell}$ between the PFC regions and the 43 cortical regions were established at the single-neuron level. The ratio of the increase in both intrahemispheric and interhemispheric connectivity densities relative to the population level was then calculated. These ratios were applied to the intrahemispheric and interhemispheric connectivity densities between the 43 regions,

yielding the adjusted intrahemispheric $\mathbf{I}_{All}^{correct}$ and interhemispheric connectivity densities $\mathbf{C}_{All}^{correct}$:

$$\mathbf{I}_{All}^{correct} = \mathbf{I}_{All} \cdot (1 + (\mathbf{I}_{cell} - \mathbf{I}_{PFC})/\mathbf{I}_{PFC}), \qquad (4)$$

$$\mathbf{C}_{All}^{correct} = \mathbf{C}_{All} \cdot (1 + (\mathbf{C}_{cell} - \mathbf{C}_{PFC})/\mathbf{C}_{PFC}). \qquad (5)$$

The Pearson correlation was computed between the time series of each pair among the 62 regions, serving as a region-to-region FC metric. The time series for each region was calculated as the average of the time series of all pixels within the respective region. The resulting FC matrix was then integrated with the structural connection matrix, retaining only those FCs where structural connections were present. To obtain region-to-region FCs under different states, the time series corresponding to those states were initially extracted, followed by repeating the previous operations to generate the analyzable FCs. For the analysis, FCs were divided into three categories: homotopic connections (interhemispheric connections between homotopic regions), interhemispheric heterotopic connections (interhemispheric connections between non-homotopic regions), and intrahemispheric heterotopic connections. To quantify the differences in FC patterns produced by different states, we perform pairwise subtraction of FC matrices corresponding to the three states. Subsequently, a *t* test was performed on the resultant difference matrices to ascertain the statistical significance of the distinctions between the states. The quantification of differences in FC patterns between intra- and interhemispheres across 3 states follows the same operation. The comparison of FC strength among various types of connections underwent Tukey's honestly significant difference (HSD) test, a two-sided test, followed by adjustment utilizing Holm's multiple comparison method.

**Stability of functional connectivity.** FC matrices were generated for each brain state. Connections with a structural basis were extracted from these matrices, reshaped into vectors, and subsequently compiled into a data matrix, **X**, ordered first by sample, then by condition. Concurrently, a state marker matrix, **Y**, was assembled with each row corresponding to a row in **X** and each column representing a distinct state. The first step of PLS analysis involved computing a covariance matrix for **X** and **Y**, illustrating the correlation between FCs and conditions. Subsequently, the covariance matrix underwent singular value decomposition (SVD), yielding multiple interpretable latent vectors (LVs). The first LV typically accounted for the largest covariance proportion. Each LV encompassed a set of saliences indicating the spatiotemporal pattern, a singular value reflecting the covariance between FCs and conditions, and a contrast between states. Thus, each LV denoted a state-associated combination of FCs, optimally weighted. Absolute salience values demonstrated the varying influence of states on FCs; larger saliences denoted higher state influence, while smaller values indicated relative state stability. The significance of each LV was evaluated through permutation testing. This involved creating 1000 permuted samples by random subject and condition label reordering within the brain set, keeping the condition set labels constant. This resulted in 1000 new covariance matrices representing the null hypothesis. Each permuted covariance matrix was subjected to SVD, yielding a null distribution of singular values. The original LV's significance was assessed by comparing its singular value to the permuted distribution. Bootstrap sampling, creating 1000 samples by resampling subjects within each condition while preserving condition labels, was used to assess the reliability of each functional connection's expression of the LV pattern. Each bootstrap sample underwent SVD, and the bootstrapped dataset's saliences were used to construct a sampling distribution of saliences from the original dataset. This process aimed to assess each salience's reliability, with broader distributions indicating saliences that highly depended on the included participants. The bootstrap distribution then served to estimate the saliences' standard errors and confidence intervals. The comparison of FC variability across various types of connections involved conducting a Tukey-HSD test, a two-sided test, subsequently applying Holm's correction for multiple comparisons.

**Statistics and reproducibility**
No statistical method was used to predetermine sample size. No data were excluded from the analyses. The experiments were not randomized. The Investigators were not blinded to allocation during experiments and outcome assessment.

**Reporting summary**
Further information on research design is available in the Nature Portfolio Reporting Summary linked to this article.

## Data availability
The Allen population tracing data are from a public database via https://connectivity.brain-map.org. The ION single-neuron tracing data are from the Brain Science Data Center of the Chinese Academy of Sciences (CEBSIT/ION Digital Brain, www.digital-brain.cn) via https://mouse.digital-brain.cn/projectome/pfc. The wide-field imaging data were deposited on ScienceDB (https://www.scidb.cn/en) and are available via dataset https://doi.org/10.57760/sciencedb.17064. Source data of all figures in the main contents and all Supplementary Figs. are provided with the paper. Source data are provided with this paper.

## Code availability
The codes used in this study are publicly available via https://github.com/marmosetbrainmapping/codes_Fei2024.

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

## Acknowledgements

We thank Prof. Linda Richards, Dr. Xiaowei Gu, and Dr. Zhe Sun for their valuable comments and suggestions on this manuscript. We thank the Brain Science Data Center of the Chinese Academy of Sciences for organizing the ION single-cell projectome data. We thank the Mouse Facility of the Center for Excellence in Brain Science and Intelligence Technology (Institute of Neuroscience) for taking care of the animals. The study was supported by grants from the National Science and Technology Innovation 2030 Major Program (Grant No. STI2030-2021ZD0203900 and 2022ZD0205000 to C.L.), the National Natural Science Foundation of China (U23A20335, 42271315, 61936007, 62036011 to S.Z., 32171088 to C.L.), and the Lingang Laboratory Grant (No. LG-QS-202201-02 to C.L.). The Guangdong Basic and Applied Basic Research Foundation (2022A1515011418 to S.Z.), Science and Technology Support Project of Guizhou Province under Grant [2021]432 to S.Z., and the Shenzhen Science and Technology Program [JCYJ20220530161409021 to S.Z.].

## Author contributions

C.L., S.Z. and J.H. supervised the study; Y.F., C.L., and Q.W. performed the data analysis and interpretation; Q.W. performed the computational modeling; K.S. collected the wide-field imaging data; C.L., Q.W., Y.F. and S.Z. wrote and edited the manuscript.

## Competing interests

The authors declare no competing interests.
