## [Peer Review File · Nature Communications]

Diverse and Asymmetric Patterns of Single-Neuron
Projectome in Regulating Interhemispheric ConnectivityReviewers' Comments:

Reviewer #1:

Remarks to the Author:

The manuscript by Fei et al., "Diverse and asymmetric patterns of single-neuron projectome in regulating interhemispheric connectivity," aims to comprehensively characterize the interhemispheric connectivity in the cortex at the structural and functional levels. On the one hand, the work presents an exhaustive and very informative re-analysis of previous and public data sets from the Allen Mouse Brain Connectivity Atlas complemented by additional single-cell reconstructions of projection patterns from another repository.

Together, these two analyses provide definitive findings extending the projection patterns' diversity much further than previously indicated. This is an essential addition to the field but not of sufficient novelty by itself. In a rightful attempt to increase the value of their findings, the authors have searched for the meaning of this connectivity at the functional level. They have investigated functional connectivity with a widefield imaging system that records with multichannel electrodes across the entire dorsal cortex, maps the activity to each cortical area (31 in total), and determines functional connectivity. They found that intrahemispheric heterotopic connections have higher functional connectivity than their interhemispheric counterparts. Finally, they establish a model to examine the properties of these connections. The model is constructed based on a well-established model working as self-organized critically, previously established. The model evaluates upstream heterotopic connectivity, both intracortical and interhemispheric, and concludes that upstream regions that project both intra and interhemispherically, are less stable.

These has been a difficult manuscript to evaluate due to the interdisciplinary approaches and the organization of the information. Although I recognize the amount of work the manuscript contains, the nexus of the structural and functional part, provided mainly by the model, are not strong and the model, as it is presented is not solid enough to solve this weakness.

Major concerns-

The manuscript is difficult to comprehend. The concepts are obscure and the rationale is difficult to follow. It is often, particularly when describing the results of the model, that the more simplistic and expected aspects are discussed as novel and of relevant, while the newer and more complex findings are just simplified. I understand that this is likely with a good intention and an attempt to make things available to a broad audience, but it is firing back and making the manuscript almost unreadable. In the part describing the functional connectivity is also very difficult to understand the link to the previous description of structural connectivity, the methodology and the findings.

Similarly, I find unfortunate the election of naming heterotopic connections only to those projecting interhemispherically, rather to those projecting to other areas, which to my knowledge, is the accepted meaning of heterotopic. The experimental methodology for the analysis of the functional connectivity and the use of the structural data for this part is also difficult to follow.

The description and discussion of the mathematical model, like other parts in the text, contains a number of inexactitudes or omissions that need to be corrected before a version of this manuscript can be published. I think it should be improved to show that the authors (i) are in full command of the mathematical model and (ii) that the narrative that they are trying to push with this paper is well connected in itself. As it is, it seems that the narrative is not fully supported by the combination of data and models employed.

The model seems to fit the experimental data but it is not challenged and the conclusions are not tested experimentally. The model is only loosely informed by the data. On the other hand, the model is very complex, with too many parameters (i.e. too many mechanisms that should be tested). This prevents us from concluding that the elements mentioned, and only those elements mentioned, are the definitive causal mechanisms of relevance. From the model results reported, we cannot rule out that communication between homotopic areas is not regulated by alternative pathways. Furthermore, the result obtained from the model that supports modulation by shared upstream inputs is rather trivial—such a complex model is not needed for that. It does not seem an ironclad story that has been tested against alternatives. The introduction of superfluous concepts does not contribute to clarify this

intricate story.

A great example of this is the mention of criticality, which the Authors make an effort to introduce (both by using a model with a critical regime and both by spending text and plots on this topic). But, if we pay attention, we can see that this concept is not relevant at all for the matter under research (i.e. to whether regulation from upstream inputs regulates communication between homotopic areas). Criticality is used, most likely unintentionally, to deliver a message that is not supported by the results, and this should be corrected.

Regarding the tangential relevance of criticality: Up to three panels are devoted to explaining critical regimes in the model (Fig. 5D, 5E, and 5F). The only relevant plot here is Fig. 5F-right. This plot shows that an increased overlap in upstream input results in an increased functional correlation between homotopic areas. In other words: Say we have two brain regions A and B that are similar (in a statistical fashion, meaning similar average connections, etc.). Now say we give each of them an input stream. Fig. 5F-right says that the more similar their input streams are, the more likely it is that A and B activity is correlated. Besides being trivial, this is true for the whole range of the model parameter plotted in Fig. 5F-right, disregarding whether the network is in a critical regime or not. The model contains parameters to control homotopic connections and other sources of inputs, but the Authors do not change these settings to showcase the effect of the different mechanisms—instead, they spend valuable time on a superfluous concept (criticality—which, nevertheless, sounds very important). Figs. 5G-H supposedly finds empirical validation of the role of shared upstream inputs, but they do so with very weak correlation coefficients and p-values and after rejecting an outlier without clear justification. Indeed, with such low correlations, it feels very bold to dismiss the role of homotopic connection given that upstream regulation is accepted.

In the final part of the section devoted to the mathematical model, the Authors move on to study what they term “divergence”. This is yet another complicated word—but it actually carries the same information as “overlap” earlier. “Divergence” measures whether the input to several (instead of two) downstream centers comes from the same set of neurons within the upstream region (high “divergence”) or whether each downstream center receives input from a disjoint subset of upstream neurons (low “divergence”). The key plot in this case is Fig. 6D. It shows that implementations of the model with no divergence (i.e. inputs from disjoint sets of neurons) achieve critical regimes for a wider range of the criticality parameter (blue curve). Instead, implementations with higher divergence (i.e. a unique subset of neurons serves as input to several downstream regions) result in criticality for a narrower range of the relevant parameter (brown curve). This finding has no bearing on the main hypotheses of the paper. However, the Authors suggest that a broader range of critical behavior entails broader stability of the critical regime, and then they proceed to connect this with the stability of connections found in earlier parts of the manuscript. How is this connection made? It seems that the term “stability” referring to two different concepts is used with ambiguity. On the one hand, the stability of critical regimes in this model has nothing to do with the stability of connections between different brain regions during different cognitive states. We are dealing with the stability of a specific dynamical regime, not with the existence and vanishing of functional connections. On the other hand, as discussed in the previous paragraph, the critical regime is not necessary at all for the results regarding functional correlation and upstream regulation.

I understand that there is novelty in the potential conclusions of these studies and relevance within the data but it should be elaborated much more to increase precision and rigourness. Perhaps it would be better to describe them separated.

Reviewer #2:

Remarks to the Author:

In this paper, the author studied homotopic and heterotopic inter-hemispheric connections. The author first employed structural tracing data to delineate interhemispheric connection patterns and then used functional imaging to uncover the structural-functional relationship. The author found a stable heterotopic connection in functional imaging and conducted a network simulation to explain its structural foundation.

Overall, I am impressed by the discovery of the important role of heterotopic connection and the computational model the author used to explain the structural-functional relationship. However, I think the idea is not delivered clearly in the current form, the sections are not closely related. Some unnecessary and confusing contents must be reorganized before the publication of this paper.

Major points:

1. The comparison between bulk tracing and ION single-neuron tracing is not convincing (see 2 below), and I don't see the necessity of including two different modalities in this research. The layer-specific analysis, morphology analysis, and degree analysis are not related to the following sections. The authors employed AAV anterograde bulk tracing data, single-neuron tracing data and conducted widefield calcium imaging. Many types of data were involved in this paper. However, the correlation of those data is not clear. The analysis is also relatively independent of each other. Are there any characteristics between interhemispheric connection patterns and functional connectivity?
2. The second major point is that it is unclear what the major conclusions from the paper actually are. The authors report more complex and densely interconnected patterns than previously acknowledged, but it is not well explained. How to understand the complexity, is there any stable and significant phenomenon? A better explanation of the conclusion obtained from those figures would help to improve the quality of this article.
3. In Figure 3, the division of states seems pointless, the FC matrixes of these states are greatly similar, and no more state-related issues were discussed in the following sections.
4. In Figures 5 and 6, the author employed the concept of criticality, but how the empirical data reveals the property of criticality is not discussed, and the relationship between criticality and functional connectivity is also unclear.

Some other more detailed comments are listed below:

1. In Section 1 (page 5), how are the injection spots selected from the database? Would the number and the distribution of selected spots influence the outcoming result, especially the analysis concerning degree calculation?
2. On page 8, lines 1-4, the connection density of Allen bulk tracing data was compared to the ION single-neuron data. According to the method, the former is calculated based on binarized intensity, and the latter is calculated based on neural number. Are these two modalities comparable in absolute value? More quantitative analysis must be made before a comparison between these two completely different modalities. The following conclusions drawn by this analysis are questionable.
3. Page 8, line 8, The authors mentioned that the distribution of subtypes in layer 1 and ACAv is different from other layers. Please explain it.
4. Fig1d, what are the layer patterns of interhemispheric connections? The patterns of different layers are basically the same. These figures do not have particularly significant patterns, so the authors may also obtain similar matrices using some unrelated data. The authors focus on the layer patterns in Fig 1, and the subsequent figures (Figs. 2-6) have nothing to do with the layer information. The relationship between these figures is unclear.
5. Fig2a, the authors claimed that brain regions appeared to be more densely connected at the single-neuron level. Whether quantitative evaluation data is available? What is the degree of density? Is it significant or not?
6. Fig 2e, the axonal morphologies of different neuronal types should be substantiated with statistical analysis. Please substantiate this statement with statistical analysis.

7. Fig 2e,f, The Y-axis in the sub-fig (the total axonal length, max axonal length, and total dendrite length) should be corrected as length(mm).

8. Page 11, line 14, The author mentioned that "The observed brain states exhibited highly similar but not identical patterns of interhemispheric functional connectivity" in fig 3. This conclusion seems unconvincing, and it doesn't make any sense. It seems there is no significant difference between intrahemispheric and interhemispheric connections. The correlation matrixes are also highly similar between the 3 states. Is it necessary to distinguish the three kinds of brain states? I would suggest performing a shuffled test or controlling experiment to validate that the correlation is not an artifact result of background or crosstalk of pixels.

9. Fig 3b, the brain activity patterns are characteristic in each brain state, while the connection map in fig3c is highly similar. It looks like the exact opposite conclusion. Moreover, the author did not mention those two sub-figs in the manuscript text. Please explain it in the main text.

10. On page 12, line 21, 'Similar results were obtained when comparing FC strengths in individual brain states.' But in Figure 4a, only in the NREM state, the heterotopic connections are significantly higher than intrahemispheric connections.

11. I am also concerned about the accuracy of brain atlas registration. The captions for A and B are incorrect as shown in fig.S6. In terms of CCF registration, how are the landmarks identified on the exposed cortex surface? Nine markers (depicted as red dots) were manually annotated on each frame, and how to accurately determine the position of these 9 points. I might have missed it but I couldn't find the information about the size of the cranial window. How to make sure the cortex is completely exposed to determine the outer edge of the cortex surface?

12. Fig4a, "The left column presents the results of functional connectivity from all anatomically connected regions, which have more intrahemispheric connections than 3 heterotopic connections." Is that correct? Again, seems the brain states have no effect on the functional connectivity.

13. For computation modeling, why the Ti-Tc overlap is set at 0.4. Does this mean that different parameter settings will lead to completely different conclusions? The author built the anatomical patterns model and neuronal network model under different types of data. It is still not clear to me what the conclusion is. Heterotopic upstream projections play a pivotal role in regulating the strength of homotopic functional connectivity. This kind of conclusion is very vague.

14. In Figure 5, I was wondering in the simulated network, is FC also correlated with Ti-Tc overlap instead of homotopic projection strength?

15. On page 17, line1-9, the author proved that lower divergence leads to criticality, but how criticality leads to lower FC variability (more robust connectivity) needs further clarification.

16. The font in all sub-figures is different (fig 5d-h, fig6). The coordinates of some diagrams are difficult to distinguish.

17. Page 17, line 6, Please keep the sentences in the same tense.

**Summary of the revision (to all reviewers):**

We appreciate your insightful comments. Through additional analyses, a new
computational model, and extensive manuscript revision, we have addressed most of
the concerns and provided a clearer and rigor elaboration of the results. Our main
revisions include the following items:

**1. Manuscript Reorganization:** We improved the manuscript's structure for better
coherence. Previously, sections on anatomical patterns, functional patterns, and
computational modeling were loosely connected. Now, we systematically progress
from identifying structural connectivity features to modeling their impact on neural
activity, and finally, to validating these models with functional data. Details with
limited relevance to the main topics have been moved to Supplementary Materials or
removed from the manuscript.

**2. Structural Data Integration:** We merged structural data from the Allen
population and ION single-neuron datasets for cohesive comparison in one figure and
one section. This involved a focused examination of intrahemispheric and
interhemispheric connectivity, revealing denser and more diverse patterns at the
single-neuron level. Degree analysis was removed, and layer-specific and neuron
morphology analyses were condensed and moved to Supplementary Material,
sharpening the paper's focus.

**3. New Computational Model:** Responding to your suggestions, we replaced the
previous model with a simpler network model. This model introduces 'heterogeneity'
as a key metric and examines its influence on neural dynamics and correlations
among brain regions. The model's predictions are corroborated using widefield
functional imaging data, offering direct insights into the structural features' impact on
functional connectivity.

**4. Functional Imaging Validation:** We have aligned the functional connectivity
analysis from widefield imaging more closely with the rest of the manuscript. This
alignment ensures that the computational modeling's predictions about the impact of
structural connectivity are validated through relevant functional data, creating a
cohesive narrative across the sections.

In summary, these revisions have significantly strengthened the manuscript, offering a
more integrated, clear, and rigorous presentation of our findings. Point-to-point
responses to each reviewer's comments are as follows:

**Responses to Reviewer #1**

*The manuscript by Fei et al., "Diverse and asymmetric patterns of single-neuron*
*projectome in regulating interhemispheric connectivity," aims to comprehensively*
*characterize the interhemispheric connectivity in the cortex at the structural and*
*functional levels. On the one hand, the work presents an exhaustive and very*
*informative re-analysis of previous and public data sets from the Allen Mouse Brain*
*Connectivity Atlas complemented by additional single-cell reconstructions of*
*projection patterns from another repository.*

*Together, these two analyses provide definitive findings extending the projection*
*patterns' diversity much further than previously indicated. This is an essential*
*addition to the field but not of sufficient novelty by itself. In a rightful attempt to*
*increase the value of their findings, the authors have searched for the meaning of this*
*connectivity at the functional level. They have investigated functional connectivity*
*with a widefield imaging system that records with multichannel electrodes across the*
*entire dorsal cortex, maps the activity to each cortical area (31 in total), and*
*determines functional connectivity. They found that intrahemispheric heterotopic*
*connections have higher functional connectivity than their interhemispheric*

*counterparts. Finally, they establish a model to examine the properties of these*
*connections. The model is constructed based on a well-established model working as*
*self-organized critically, previously established. The model evaluates upstream*
*heterotopic connectivity, both intracortical and interhemispheric, and concludes that*
*upstream regions that project both intra and interhemispherically, are less stable.*
*These has been a difficult manuscript to evaluate due to the interdisciplinary*
*approaches and the organization of the information. Although I recognize the amount*
*of work the manuscript contains, the nexus of the structural and functional part,*
*provided mainly by the model, are not strong and the model, as it is presented is not*
*solid enough to solve this weakness.*

**General response:** At the beginning of your comment, you provided an accurate
summary of our work. We appreciate it very much. Your important and insightful
critiques and questions greatly helped us enhance the quality of the article. The
following are our responses to your comments.

*Major concerns-*

*The manuscript is difficult to comprehend. The concepts are obscure and the rationale*
*is difficult to follow. It is often, particularly when describing the results of the model,*
*that the more simplistic and expected aspects are discussed as novel and of relevant,*
*while the newer and more complex findings are just simplified. I understand that this*
*is likely with a good intention and an attempt to make things available to a broad*
*audience, but it is firing back and making the manuscript almost unreadable.*

**Response #1:** We are sorry that the original manuscript was difficult to follow. In the
revised manuscript, we improved the organization and the logic of the manuscript (see
**Response #2 to Reviewer #1**) and adopted a more direct and simpler model to
explore the functional implication of the projection patterns (see **Response #4 to**
**Reviewer #1**). We also provided a more detailed and computational description of the
new model in the revised methods. Thus, the revised manuscript is easy to follow for

both computational scientists and broad audiences.

*In the part describing the functional connectivity is also very difficult to understand*
*the link to the previous description of structural connectivity, the methodology and the*
*findings.*

**Response #2:** We concur with you that our elucidation of the link between structural
and functional analysis was unclear in the original manuscript. Therefore, in the
updated manuscript, we reorganized the article as follows: we first identified features
from the structural connectivity, then modeled their predicted impacts on neural
activity using new models based on your suggestions, and finally validated these
predictions through analysis of functional data. Specifically, the structural feature
"heterogeneity" was extracted from the single-neuron structural connectivity. Via
modeling, we predicted heterogeneity to affect the correlations between brain regions.
These predictions were validated by experimentally measuring the correlations using
the wide-field imaging data, namely calculating the functional connectivity between
regions.

*Similarly, I find unfortunate the election of naming heterotopic connections only to*
*those projecting interhemispherically, rather to those projecting to other areas, which*
*to my knowledge, is the accepted meaning of heterotopic.*

**Response #3:** We adopted the names from a previous study highly related to our
topic¹ for a consistency purpose, but we agreed that their names were not accurate as
you suggested. In the revised manuscript, we used more specific terms to describe
these connection types: "homotopic (homo)", "intrahemispheric-heterotopic
(intra-hete)", and "interhemispheric-heterotopic (inter-hete)". The "heterotopic"
included both intrahemispheric and interhemispheric connections. The
"interhemispheric (or contralateral)" included both "homotopic (homo)" and
"interhemispheric-heterotopic (inter-hete)". The "intrahemispheric (or ipsilateral)"

only included intrahemispheric-heterotopic (page 5: 108-110).

*The experimental methodology for the analysis of the functional connectivity and the*
*use of the structural data for this part is also difficult to follow.*

*The description and discussion of the mathematical model, like other parts in the text,*
*contains a number of inaccuracies or omissions that need to be corrected before a*
*version of this manuscript can be published. I think it should be improved to show that*
*the authors (i) are in full command of the mathematical model and (ii) that the*
*narrative that they are trying to push with this paper is well connected in itself. As it*
*is, it seems that the narrative is not fully supported by the combination of data and*
*models employed. The model seems to fit the experimental data but it is not*
*challenged and the conclusions are not tested experimentally. The model is only*
*loosely informed by the data.*

**Response #4:** Following the suggestions, we replaced the original model of
self-organized criticality with a simpler and more direct network model with sparse
and balanced excitatory and inhibitory connections, which captured the mean
behavior of many widely used neuron network models of excitatory and inhibitory
interactions like the Wilson-Cowan model². Unlike the previous model in which there
were complex interactions between the spiking activity and synaptic strength, the new
model considered the synaptic strength to be static.

The structural feature " T_i - T_c overlap" used in the previous manuscript was replaced
with the new term "heterogeneity". We use heterogeneity to measure the level of
non-overlap of the projections from a specific upstream region (U) to downstream
regions (D), both for the projections to a pair of homotopic regions (**Fig. R1A**) and
for projections to multiple heterotopic regions (**Fig. R1B**). Then we used the model to
examine how heterogeneity of projections regulated the 1) the correlation between the
neural dynamics of homotopic downstream pairs, and 2) the correlation among
heterotopic downstream regions (**Fig. R1C-J**). Finally, we used the widefield

functional imaging data to verify the prediction of the network modeling on the
heterogeneity (**Fig. R1K-L**). The quantification of heterogeneity was described in the
revised manuscript result part **Quantification of Projection Heterogeneity (page 9**
**line 188 - page 11 line 224)**. The modelling results were described in the revised
manuscript result part **Computationally Modeling the Impact of Projection**
**Heterogeneity on Neural Dynamics (page 11 line 228 - page 15 line 316)**. The
experimental validations were described in the revised manuscript result part
**Strength and Stability of Interhemispheric Functional Connectivity (page 15 line**
**318 - page 19 line 396)**.

In brief, for the modeling, we simulated interactions among numerous brain regions to
examine how regional dynamics changed due to projection heterogeneity. Our model
incorporated hundreds of neurons within each brain region, interconnected by
neuron-to-neuron projections (**Fig. R1C, Fig. 3A, Fig. 4A**). We specifically explored
two scenarios: 1) projections targeting homotopic regions, and 2) projections to
heterotopic regions. In the former, we adjusted the model's heterogeneity by varying
the overlap among subsets of neurons (**Fig. R1D, Fig. 3B, page 11 line 242 - page 12**
**line 245**). Our findings indicated that homotopic functional connectivity strength
diminished as heterogeneity increased (**Fig. R1E, Fig. 3C, page 12: 260-262**). For
heterotopic projections, we introduced a single parameter, ' h ', to smoothly modulate
neuron group overlaps, thereby controlling heterogeneity (**Fig. R1F, Fig. 4B-C, page**
**14: 290-295**). We initially observed that heterogeneity impacted the network
dynamics' overall stability, evidenced by fewer state transitions with increased
heterogeneity (**Fig. R1G-H, Fig. 4D-E, page 14: 297-305**). Furthermore, we analyzed
functional connectivity strength and variability, measured respectively as the average
and variance of inter-regional correlation coefficients over time. The results suggested
that while heterogeneity decreased the strength of inter-regional functional
connectivity, it simultaneously lowered its variability, enhancing stability (**Fig. R1I-J,**
**Fig. 4F-G, page 14 line 307 - page 15 line 316**).

Validation of the model was achieved using widefield functional imaging data. For
homotopic downstream pairs, a negative and significant correlation was observed
between homotopic functional connectivity and the heterogeneity of the upstream
projections, supporting our model's results (**Fig. R1K, Fig. 6A, page 16: 332-344**).
Additionally, for ipsilateral (intrahemispheric-heterotopic) connections, both
functional connectivity strength and variability were significantly higher compared to
contralateral (interhemispheric-heterotopic) connections. This aligned with our
model's predictions, as ipsilateral projections demonstrated lower heterogeneity in
single-neuron data, suggesting stronger and more variable (less stable) functional
connectivity than contralateral connections (**Fig. R1L, Fig. 6C-D, page 16 line 349 -**
**page 18 line 385**).

Overall, the revised model and manuscript presented a more coherent and
comprehensible structure. We also enhanced the manuscript by providing an extensive
description of the mathematical model in the revised methods section.

**Figure R1. Definition of heterogeneity and the new neuronal network model.** (A) The
 heterogeneity of projections to a pair of homotopic regions (u_i & u_c , from U to D_i & D_c) is
 calculated as one minus the ratio of overlap of u_i and u_c . (B) The heterogeneity of projections to
 multiple heterotopic regions is represented by the distribution function of the number of
 downstream projecting neurons. Different levels of non-overlap results in different heterogeneity
 (upper), indicated by the different shapes of the distribution function (lower). (C) The illustration
 of the model for investigating the heterogeneity of two types of projections. Left: the model for
 the projections to a pair of homotopic regions. Right: the model for projections to multiple
 heterotopic regions. (D) For projections to a pair of homotopic regions, heterogeneity is tuned by

changing the overlap of neuron subsets in the model. **(E)** In the model result, the correlation
between activities of homotopic regions will decrease with the increase of heterogeneity. **(F)** For
projections to multiple heterotopic regions, heterogeneity is tuned by a single parameter h that
regulates the overlap of multiple neuron subsets (upper). When h increases (color from yellow to
blue), the distribution function is more concentrated to $N=1$, indicating increased heterogeneity.
**(G-H)** In the model result, heterogeneity affects the state transition behavior (between
asynchronous and synchronous states, G) of the network dynamics, with the duration of the
synchronous periods decreasing with the increase of heterogeneity (H). **(I-J)** In the model result,
the strength of correlation (I) and the variability of correlation (J) both reduce when heterogeneity
gets larger. **(K)** Validating model predictions for heterogeneity of projections to a pair of
homotopic regions. Widefield data showed that the correlation between homotopic regions is
negatively correlated with heterogeneity, with heterogeneity calculated from the single-neuron
data. **(L)** Validating model predictions for heterogeneity of projections to multiple heterotopic
regions. Widefield data showed that the strength and variability of correlation between ipsilateral
regions are stronger than that between contralateral regions.

*On the other hand, the model is very complex, with too many parameters (i.e. too*
*many mechanisms that should be tested). This prevents us from concluding that the*
*elements mentioned, and only those elements mentioned, are the definitive causal*
*mechanisms of relevance. From the model results reported, we cannot rule out that*
*communication between homotopic areas is not regulated by alternative pathways.*
*Furthermore, the result obtained from the model that supports modulation by shared*
*upstream inputs is rather trivial—such a complex model is not needed for that.*

**Response #5:** You were concerned that there were other parameters besides T_i-T_c
overlap that could potentially affect the correlation between activities of homotopic
regions. Also, you pointed out that the result of the original complex model was trivial.
We thank you for these insightful comments. In the revised manuscript, we performed
a comparison of different pathways that may regulate the homotopic communication
in the model (**Fig. R2A, Fig. 3B, page 11 line 242 - page 12 line 258**), including

heterogeneity (the feature we used to represent non-overlapping in the revised
manuscript, substituting " T_i-T_c overlap"), homotopic strength (the probability of direct
homotopic projections), and self-homotopic overlap (overlap of self-projections and
homotopic projections of neurons in the upstream). The last two parameters
represented the strength of direct connections between homotopic regions. In the
model result, changes in homotopic strength and self-homotopic overlap affected the
direct structural connections between homotopic regions, which were traditionally
viewed as key determinants of homotopic correlation. Consequently, our study aimed
to identify which of these three structural features exerted the most significant
influence.

Through network modeling, we found that the homotopic correlations were negatively
correlated with heterogeneity and positively correlated with self-homotopic overlap
and homotopic strength (**Fig. R2B-D, Fig. 3C-E, page 12 line 260 - page 12 line**
**262**). Interestingly, the range of variation in correlation values attributed to
heterogeneity exceeded those due to homotopic strength and self-homotopic overlap
(**Fig. R2B-D, Fig. 3C-E, page 12 line 262 - page 13 line 271**). This might be a
consequence of the relative abundance of heterotopic projections compared to
homotopic ones; specifically, in a network with $N=10$ regions, each region received
$(N-1)\times 2=18$ heterotopic projections versus a single homotopic projection. To further
probe this phenomenon, we analyzed a minimal network model with just two pairs of
homotopic regions ($N=2$), where the number of homotopic projections mostly
approximated that of heterotopic projections. In this minimal model, the variation
range caused by self-homotopic overlap and homotopic strength was comparable to
that caused by heterogeneity (**Fig. R2B-D, Fig. 3C-E, page 12 line 262 - page 13**
**line 271**). Furthermore, the differences between the variation range caused by
heterogeneity and homotopic strength or self-homotopic overlap, denoted as
$\Delta(\text{variation range})$, kept expanding as the number of regions N in the model increased
(**Fig. R2E-F, Fig. 3F-G, page 13: 272-279**). This suggested that the impact of
heterogeneity became more pronounced with the increase of projection heterogeneity

in the network. Considering the mouse cortex, which comprised numerous
interconnected brain regions (N was large), we can infer that the heterogeneity of
upstream projections was the dominant factor in shaping homotopic correlation,
outweighing the effects of direct connections between homotopic regions.

Furthermore, the prediction was consistent with the experimental data. We examined
the correlations between homotopic FC and both heterogeneity and homotopic
connection strength using our wide-field imaging data. Our findings revealed a
predominantly negative correlation between homotopic FC and heterogeneity ($r =$
10 -0.626 , $p = 0$) but a weak non-significant correlation between homotopic FC and the
11 strength of homotopic connections ($r = 0.285$, $p = 0.12$) (**Fig. R2G-H, Fig. 6A-B,**
**page 16: 332-346**). These observations supported our model's prediction that
heterogeneity of upstream projections was a key factor in influencing homotopic
correlation.

1

2 **Figure R2. Network dynamics modeling of homotopic region pairs.** (A) Three structural
 3 features were investigated in the model: heterogeneity, homotopic strength, and self-homotopic
 4 overlap. (B-D) All three features affect the homotopic correlation. When the number of homotopic

region pairs, is large ($N=10$, upper), the variation range between the minimum and maximum
correlation values caused by heterogeneity is the largest; when $N=2$ (bottom), the variation ranges
caused by three features are comparable. **(E, F)** The difference between variation ranges caused by
heterogeneity and homotopic strength (E) or heterogeneity and self-homotopic overlap (F) will
increase with the increase of N . **(G, H)** Widefield data show that homotopic correlation is
negatively and significantly correlated with heterogeneity, but has no significant correlation with
homotopic connection strength.

*It does not seem an ironclad story that has been tested against alternatives. The*
*introduction of superfluous concepts does not contribute to clarify this intricate story.*

*A great example of this is the mention of criticality, which the Authors make an effort*
*to introduce (both by using a model with a critical regime and both by spending text*
*and plots on this topic). But, if we pay attention, we can see that this concept is not*
*relevant at all for the matter under research (i.e. to whether regulation from upstream*
*inputs regulates communication between homotopic areas). Criticality is used, most*
*likely unintentionally, to deliver a message that is not supported by the results, and*
*this should be corrected.*

*Regarding the tangential relevance of criticality: Up to three panels are devoted to*
*explaining critical regimes in the model (Fig. 5D, 5E, and 5F). The only relevant plot*
*here is Fig. 5F-right. This plot shows that an increased overlap in upstream input*
*results in an increased functional correlation between homotopic areas. In other*
*words: Say we have two brain regions A and B that are similar (in a statistical fashion,*
*meaning similar average connections, etc.). Now say we give each of them an input*
*stream. Fig. 5F-right says that the more similar their input streams are, the more*
*likely it is that A and B activity is correlated. Besides being trivial, this is true for the*
*whole range of the model parameter plotted in Fig. 5F-right, disregarding whether*
*the network is in a critical regime or not. The model contains parameters to control*
*homotopic connections and other sources of inputs, but the Authors do not change*
*these settings to showcase the effect of the different mechanisms—instead, they spend*
*valuable time on a superfluous concept (criticality—which, nevertheless, sounds very*

*important).*

**Response #6:** The concept of criticality was indeed not enough relevant for the other
parts of the results. Following your suggestions, we have abandoned the concept and
applied more relevant model indexes: the strength and variability of the Pearson
correlation coefficients between brain regions' simulated activities. Specifically, the
strength of correlation is calculated as the mean correlation value over time, and the
variability of correlation is calculated as the standard deviation. The new modeling is
described in the **Response #4-#5 to Reviewer #1.**

*Figs. 5G-H supposedly finds empirical validation of the role of shared upstream*
*inputs, but they do so with very weak correlation coefficients and p-values and after*
*rejecting an outlier without clear justification. Indeed, with such low correlations, it*
*feels very bold to dismiss the role of homotopic connection given that upstream*
*regulation is accepted.*

**Response #7:** In this part, we aimed to experimentally compare heterogeneity
(corresponding to T_i-T_c overlap in the original manuscript) and direct connection
strength in terms of their correlation with homotopic FC. In the previous result, the
contrast is obvious in that T_i-T_c overlap had a statistically significant linear correlation
with homotopic FC, while direct connection strength does not. Plus, we further
supplemented the analysis with fMRI data in the original manuscript. The result was
validated by fMRI data, where the correlation between homotopic functional
connectivity and T_i-T_c overlap was nearly 0.5, with p-value=0.008. In the original
manuscript, we rejected an outlier region, VISa, in analyzing the correlation between
T_i-T_c overlap and homotopic FC because the overlap level of projections to VISa was
unusually large compared with the overlap level of all other regions.

However, in the revised manuscript, we abandoned original definition overlap level.

Instead, we calculated the heterogeneity of projections as one minus a slightly
different overlap measurement, which had a simpler definition than the original one
and completely matched the definition of the corresponding parameter in modeling
(see **Methods**). No outlier was detected in the updated manuscript, and the correlation
between heterogeneity and homotopic FC was shown to be negative ($r = -0.626$) and
significant ($p = 0$). The comparison between heterogeneity and direct connections was
described in in the **Response #4-5 to Reviewer #1 (Fig. R2G-H, Fig. 6A-B, page 16:**
**332-346)**. This result aligned with the previous result, in that higher heterogeneity
indicates lower T_i-T_c overlap, leading to lower homotopic FC.

*In the final part of the section devoted to the mathematical model, the Authors move*
*on to study what they term “divergence”. This is yet another complicated word—but it*
*actually carries the same information as “overlap” earlier. “Divergence” measures*
*whether the input to several (instead of two) downstream centers comes from the same*
*set of neurons within the upstream region (high “divergence”) or whether each*
*downstream center receives input from a disjoint subset of upstream neurons (low*
*“divergence”). The key plot in this case is Fig. 6D. It shows that implementations of*
*the model with no divergence (i.e. inputs from disjoint sets of neurons) achieve critical*
*regimes for a wider range of the criticality parameter (blue curve). Instead,*
*implementations with higher divergence (i.e. a unique subset of neurons serves as*
*input to several downstream regions) result in criticality for a narrower range of the*
*relevant parameter (brown curve). This finding has no bearing on the main*
*hypotheses of the paper.*

**Response #8:** In the updated manuscript, the terms "overlap" and "divergence" were
replaced by the single term "heterogeneity". Higher overlap level, and lower
divergence level, were uniformly defined as being less heterogeneous, which meant
that the multiple neuron subsets projecting to different downstream regions had more
common neurons. The definition of heterogeneity for both the projections to a pair of
homotopic regions and projections to multiple heterotopic regions was described in

the **Response #4-5** to **Reviewer #1**. See **Methods Heterogeneity of projections to**
**pairs of homotopic regions** and **Heterogeneity of projections to mutually**
**heterotopic regions** for more details.

*However, the Authors suggest that a broader range of critical behavior entails*
*broader stability of the critical regime, and then they proceed to connect this with the*
*stability of connections found in earlier parts of the manuscript. How is this*
*connection made? It seems that the term “stability” referring to two different*
*concepts is used with ambiguity. On the one hand, the stability of critical regimes in*
*this model has nothing to do with the stability of connections between different brain*
*regions during different cognitive states. We are dealing with the stability of a specific*
*dynamical regime, not with the existence and vanishing of functional connections. On*
*the other hand, as discussed in the previous paragraph, the critical regime is not*
*necessary at all for the results regarding functional correlation and upstream*
*regulation. I understand that there is novelty in the potential conclusions of these*
*studies and relevance within the data but it should be elaborated much more to*
*increase precision and rigourness. Perhaps it would be better to describe them*
*separated.*

**Response #8:** We agree that the connection of the stability of criticality with the
stability of functional connectivity was inappropriate. We solved this problem by
adopting better model indexes (see **Response #4-#5** to **Reviewer #1**) in the new
model. Firstly, we calculated the mean inter-regional correlation under perturbations,
corresponding to the strength of functional connectivity. Secondly, we calculated the
standard deviation of inter-regional correlation, which represented the variability of
functional connectivity. In the result of our new modeling, higher heterogeneity
caused lower mean inter-regional correlation, together with lower standard deviation,
indicating that heterogeneity would reduce the strength but increase the stability of
correlation (**Fig. R1G-J, Fig. 4D-G, page 14 line 297- page 15 line 316**). This was in
accord with the result of wide-field imaging analysis that intrahemispheric correlation

was larger but less stable, with intrahemispheric connections showing lower
heterogeneity (Fig. R1L, Fig. 6C-D, page 16 line 349 - page 18 line 385).

**Responses to Reviewer #2**

*In this paper, the author studied homotopic and heterotopic inter-hemispheric*
*connections. The author first employed structural tracing data to delineate*
*interhemispheric connection patterns and then used functional imaging to uncover the*
*structural-functional relationship. The author found a stable heterotopic connection*
*in functional imaging and conducted a network simulation to explain its structural*
*foundation.*

*Overall, I am impressed by the discovery of the important role of heterotopic*
*connection and the computational model the author used to explain the*
*structural-functional relationship. However, I think the idea is not delivered clearly in*
*the current form, the sections are not closely related. Some unnecessary and*
*confusing contents must be reorganized before the publication of this paper.*

**Response #1:** We greatly appreciate your thorough examination of our manuscript.
Their constructive feedback highlighted key issues and offered detailed, insightful
suggestions for improvement. In response, we have reorganized and revised the
manuscript to address these points. Below are our specific responses to your
comments.

*Major Comments*

*1. The comparison between bulk tracing and ION single-neuron tracing is not*
*convincing (see 2 below), and I don't see the necessity of including two different*
*modalities in this research. The layer-specific analysis, morphology analysis, and*
*degree analysis are not related to the following sections. The authors employed AAV*

*anterograde bulk tracing data, single-neuron tracing data and conducted widefield*
*calcium imaging. Many types of data were involved in this paper. However, the*
*correlation of those data is not clear. The analysis is also relatively independent of*
*each other. Are there any characteristics between interhemispheric connection*
*patterns and functional connectivity?*

**Response #2:** We appreciate your insightful feedback regarding the integration and
relevance of different modalities in our study. Originally, our manuscript aimed to
provide a comprehensive view of the anatomical patterns of interhemispheric
connections, encompassing both population-level and single-neuron perspectives.
However, we acknowledge your concerns about the apparent disconnection between
these sections.

In response, we have revised our manuscript to strengthen the coherence and
relevance of these sections. Specifically:

**1. Integration of Figures:** We merged the initial Figures 1 (Allen population tracing)
and 2 (single neuron tracing) into a new Figure (**Fig. R3, Fig. 1**). This revised figure
more effectively emphasized the comparison between population-level and
single-neuron tracing patterns.

**2. Computational Modeling:** We introduced new computational models that utilized
the patterns revealed in **Fig. R1A-B, Fig. 2** (also see **Response #4-#5 to Reviewer**
**#1**). These models were designed to explore the potential impact of anatomical
patterns on functional connectivity.

**3. Validation with Wide-Field Calcium Imaging Data:** We employed wide-field
calcium imaging data to validate our computational models' predictions regarding the
influence of anatomical patterns on functional connectivity.

**4. Refinement of Anatomical Features:** Regarding the layer-specific, degree, and
morphological analyses initially presented, we took steps to refine their inclusion:

(1) Degree Analyses: We removed these from the main text due to their limited
direct relevance to our core findings.

(2) Supplementary Materials: The layer-specific and morphological analyses,
while less central to our main argument, offer valuable insights into the
structural aspects of interhemispheric connections. Therefore, we relocated
these sections to the supplementary materials, ensuring they support, rather
than distract from, the primary narrative of the paper.

(3) Morphological Evidence: We maintained the examination of morphological
properties in the Supplementary Materials, highlighting that different neuron
types (IB, IBC, BC, IC, B) exhibit diverse axonal and dendritic properties.
This diversity underscores our conclusion that single-neuron data can reveal
more varied projections, enriching our understanding of interhemispheric
connections from a structural standpoint.

Through these revisions, we believe we have addressed the concerns raised,
enhancing the manuscript's coherence and strengthening the connection between the
different data types and analyses used in our study.

 **Figure R3. Diverse patterns of interhemispheric connections.** (A) Two dorsal-view cortical flat
 maps from the CCFv3, showing 43 cortical regions and injection sites in various mouse strains
 from the Allen Mouse Connectivity Atlas (Allen population). Refer to **Tables S1**, and **S2** for strain
 and experimental details, and **Table S3** for full brain region names. The third panel depicts the

injection sites of ION single-neuron data. **(B)** Illustrations of the connection types, including
intrahemispheric-heterotopic, interhemispheric-heterotopic, and homotopic connections. **(C)**
Connectivity matrices of wild-type mice from Allen population data, with layer-specific Cre mice
data presented in **Fig. S1C** and **Fig. S1E**. Different layers generated similar intra-hemispheric
connectivity matrices, yet exhibited divergent inter-hemispheric connectivity matrices (**Fig. S1F**).
**(D)** Weighted (upper two rows) and binary (lower two rows) connectivity matrices of the PFC
projectome based on ION single-neuron data and Allen population data, respectively. **(E)**
Projection types (left) and their derived seven neuron/region types, with a gray background
indicating five types projecting to both hemispheres. **(F)** Composition of regional types in the
Allen population data (top) and neuronal types in the ION single-neuron data (bottom). **(G)**
Laminar distribution of neuronal types from ION single-neuron data. **(H)** Regional compositions
of different neuronal types from ION single-neuron data.

*2. The second major point is that it is unclear what the major conclusions from the*
*paper actually are. The authors report more complex and densely interconnected*
*patterns than previously acknowledged, but it is not well explained. How to*
*understand the complexity, is there any stable and significant phenomenon? A better*
*explanation of the conclusion obtained from those figures would help to improve the*
*quality of this article.*

**Response #3:** We understand your concern regarding the clarity of our major
conclusions and their implications. In response, we have revised our manuscript to
better elucidate these aspects, particularly focusing on the concept of 'heterogeneity'
in neuronal projections and its functional significance. We introduced 'heterogeneity'
as a quantitative metric to assess the distinctness of neuronal projections from an
upstream region to multiple downstream regions. This metric effectively captured the
degree of non-overlap among neuron groups projecting to different targets, offering a
novel perspective on the complexity of neural circuitry.

Our revised analysis, presented in **Figures R1&R2** and **Figures 3&4**, more clearly
demonstrated how this heterogeneity influences cortical dynamics by computational
modeling. The key findings from our neuron-tracing data are twofold:

• We found that projections with greater heterogeneity (i.e., less overlap among
projecting neuron groups) led to an increased correlation between activities in
homotopic downstream regions. This suggested that the specificity of
projections played a crucial role in synchronizing neural activities across
corresponding areas of the brain.
- • Our analysis also revealed that projections with higher heterogeneity to
mutually heterotopic regions resulted in weaker but more stable functional
connectivity. This finding was particularly noteworthy as it highlighted a
previously underappreciated aspect of brain organization – that greater
projection specificity can confer enhanced stability in neural correlations, even
amidst reduced connection strength.

Both of these points were strongly supported by our wide-field imaging data. These
results underscored the significant role of projection heterogeneity in shaping the
dynamics and stability of cortical networks. We believe these revisions and
clarifications directly address your concerns and significantly enhance the
manuscript's contribution to understanding the complex patterns of neural projections
and their functional implications.

*3. In Figure 3, the division of states seems pointless, the FC matrixes of these states*
*are greatly similar, and no more state-related issues were discussed in the following*
*sections.*

**Response #4:** The functional connectivity (FC) patterns across different brain
states—awake, REM, and non-REM—were visually similar in terms of the brain
regions involved (**Fig. R4B**). However, our detailed quantitative analysis uncovered

critical differences in the strength of these connections, which were not immediately
apparent from a visual inspection of the FC matrices.

To elucidate these differences, we performed a pairwise subtraction of the FC
matrices corresponding to each of the three states. Our results, illustrated in **Fig. R4C**
and **Fig. 5C**, revealed significant variations in the strength of numerous connections,
particularly when comparing the awake and REM states to the non-REM state ($p <$
0.05 , **page 15: 327-330**).

These findings were not just quantitative variations but were substantial functional
significance. The differing strengths of connections across states indicated that while
the overall connectivity pattern remained consistent, the intensity of inter-regional
communication within the brain vary considerably. This variation in connectivity
strength was pivotal in understanding how the brain modulated its information
processing and integration in different states of consciousness.

The variability in connectivity strength across different brain states also served as a
valuable metric for evaluating the predictions of our computational models
concerning the stability and dynamics of functional connectivity.

**Figure R4. Wide-field calcium imaging of different brain states. (A)** Brain activity patterns

during awake, REM, and NREM sleep states. For each brain state (each row), the left displays

examples of EEG (upper) and EMG data (bottom), with the duration of the brain state highlighted

in green. The right illustrates brain activity patterns, represented by the mean fluorescence

intensity of calcium imaging. The baseline (F) is set as the median fluorescence intensity over the

entire recording time, and the color scale displays $\Delta F/F$ values. **(B)** The three matrices depict the

functional connectivity of intrahemispheric connections, interhemispheric connections, and the

difference between intrahemispheric and interhemispheric connections. **(C)** Disparity matrices

illustrate differences in FC strength for intra- and interhemispheric connections across various
brain states. Areas shaded in gray represent connections where the FC strength differential is not
statistically significant ($p > 0.05$).

*4. In Figures 5 and 6, the author employed the concept of criticality, but how the*
*empirical data reveals the property of criticality is not discussed, and the relationship*
*between criticality and functional connectivity is also unclear.*

**Response #5:** We agree that criticality may not be the most suitable metric for our
specific research objectives. Consequently, we have revised our manuscript to exclude
the sections pertaining to criticality and the associated modeling aspects.

In place of the previous model, we have introduced a simpler, yet more direct and
relevant modeling approach. Detailed information about this revised modeling
approach can be found in **Response #4-#5 to Reviewer #1**.

*Some other more detailed Comments*

*1. In Section 1 (page 5), how are the injection spots selected from the database?*
*Would the number and the distribution of selected spots influence the outcoming*
*result, especially the analysis concerning degree calculation?*

**Response #6:** In our study, the term 'injection spots' refers to the specific sites within
the cortex where a virus was injected in various experiments. These spots denoted not
just the physical locations but also the density of injections within those experiments.
For experiments involving wild-type mice, we utilized data from publicly available
datasets, where injections were directly administered into the cortex. This data was
accessed through a designated download interface.

For experiments involving transgenic mice, specifically targeting certain cortical
layers, we initially extracted all relevant datasets from the database. These datasets
were from experiments where injections were made in the cortex region of transgenic
mice. We then refined our selection based on information from prior publications³,
focusing on specific genes of interest. This targeted approach allowed us to isolate
experimental data pertinent to our research objectives. We have included
comprehensive details of the experimental data used—including IDs, injection regions,
and corresponding transgenes—in the newly added **Table S2**, which encompasses
both wild-type and layer-specific data.

To assess the impact of the distribution of injection sites on our results, we performed
a correlation analysis between this distribution and our findings, including degree and
connection density. Our analysis indicated that there was a minimal correlation
between the distribution of injection sites and the resulting data. Specifically, the
correlation coefficients for intra-hemispheric connection density and interhemispheric
connection density in relation to the percentage of 43 cortical regions covered by the
injection sites were 0.0419 and -0.1500, respectively. Additionally, we observed a low
correlation between the degree of a region and its representation percentage in the
experimental datasets, both for intra-hemispheric and inter-hemispheric connections
(**Fig. R5**).

These findings suggested that the selection and distribution of injection sites in our
experiments had a negligible impact on the outcome of our degree calculations and
connection density analyses.

**Figure R5. Correlation between Regional Connection Degree and Tracer Injection**
 **Distribution.** These scatter plots show the relationship between regional connection degree,
 defined as the number of connections per region, and the proportion of tracer injections involving
 each region relative to the total number of injections. Data are categorized into intrahemispheric
 (left) and interhemispheric (right) connections. Each point represents an individual brain region.

*2. On page 8, lines 1-4, the connection density of Allen bulk tracing data was*
 *compared to the ION single-neuron data. According to the method, the former is*
 *calculated based on binarized intensity, and the latter is calculated based on neural*
 *number. Are these two modalities comparable in absolute value? More quantitative*
 *analysis must be made before a comparison between these two completely different*
 *modalities. The following conclusions drawn by this analysis are questionable.*

**Response #7:** We can compare the two modalities for several reasons. First, the Allen
 bulk tracing data used normalized connection strength, with a threshold defined by a
 previous study³ to binarize the matrix and determine connected regions. This
 threshold aimed to balance false negatives and positives: higher thresholds led to
 more false negatives, while lower thresholds resulted in more false positives. The
 previous study used 'single-neuron tracing' as a gold standard to set the optimal
 threshold for the Allen bulk tracing data. In our study, we applied a similar threshold
 to binarize the Allen data. Since this threshold was based on 'single-neuron tracing'

data, the resultant binarized matrix was comparable with our ION single-neuron data
in terms of which regions were connected.

Second, connection density was calculated as the ratio of the number of existing
connections to the total possible connections, focusing on the presence or absence of
connections. After thresholding, the Allen tracing data conveyed the same information
as the ION single-neuron data regarding connection presence.

Third, an important aspect of our study was the comparison of intrahemispheric and
interhemispheric connection densities. These were relative values; thus, different
thresholds would not alter the conclusions drawn from these comparisons. Moreover,
we conducted additional statistical analyses on this comparison. We calculated the
intrahemispheric and interhemispheric connection densities across various layers,
revealing significantly higher connection densities in the single-neuron data than in
the Allen population tracing data for both intra- and inter-hemispheric connections
(intrahemisphere: $p = 0.02$, interhemisphere: $p = 0.01$, **page 7: 138-140**).

*3. Page 8, line 8, The authors mentioned that the distribution of subtypes in layer 1*
*and ACAv is different from other layers. Please explain it.*

**Response #8:** Contrary to the common pattern found in other layers—where the IBC
subtype was predominant, followed by IB, BC, B, and IC—our data revealed that in
layer 1 and the ACAv region, the IB subtype was the most abundant. This deviation
from the expected distribution pattern suggested that layer 1 and the ACAv region
may have specialized functions or connectivity that necessitated a different subtype
composition.

*4. Fig1d, what are the layer patterns of interhemispheric connections? The patterns of*
*different layers are basically the same. These figures do not have particularly*
*significant patterns, so the authors may also obtain similar matrices using some*

*unrelated data. The authors focus on the layer patterns in Fig 1, and the subsequent*
 *figures (Figs. 2-6) have nothing to do with the layer information. The relationship*
 *between these figures is unclear.*

 **Response #9:** In the original Figure 1d, the matrices for intra-hemispheric and
 inter-hemispheric connections were calculated separately for each layer, and the
 differential contrast matrix was derived by subtracting inter-hemispheric from
 intra-hemispheric connections. To quantitatively assess the relationship between these
 matrices, we calculated Pearson correlations for intra- and inter-hemispheric
 connections, as well as for the contrast matrices (**Fig. R6, Fig. S1F**). Our findings
 indicated a lower correlation amongst inter-hemispheric connectivity matrices as
 opposed to a higher correlation within intra-hemispheric connectivity across layers.
 Additionally, the analysis of regional degree and strength revealed variability across
 different layers.

 Originally, our manuscript aimed to provide a comprehensive view of the anatomical
 patterns of interhemispheric connections, encompassing both population-level and
 single-neuron perspectives. However, we acknowledged your concerns about the
 apparent disconnection between these sections. In our revision, we have removed the
 degree analysis and moved most of the layer analysis into the Supplementary Material,
 ensuring they support, rather than distract from, the primary narrative of the paper.

 **Figure R6. The correlation among the connectivity matrices of different layers.** From left to
 right are the matrix of intrahemispheric connections, the matrix of interhemispheric connections

and the contrast matrix (intra-hemispheric - inter-hemispheric).

*5. Fig2a, the authors claimed that brain regions appeared to be more densely*
*connected at the single-neuron level. Whether quantitative evaluation data is*
*available? What is the degree of density? Is it significant or not?*

**Response #10:** We compared the connectivity density from prefrontal cortex (PFC)
regions to other regions at both the single-neuron and population levels. The
connectivity density was calculated as the ratio of observed connections to the
possible maximum connections. In the revised manuscript (**page 7: 138-140**), we
included a statistical comparison of the connection densities from both single-cell and
population data. Using t-tests, we found a significantly higher density at the
single-neuron level for both intra-hemispheric ($p = 0.02$) and inter-hemispheric
connections ($p = 0.01$). These results supported our claim of a denser connection
pattern at the single-neuron level.

*6. Fig 2e, the axonal morphologies of different neuronal types should be substantiated*
*with statistical analysis. Please substantiate this statement with statistical analysis.*

**Response #11:** In the revised manuscript, we performed the Tukey Honestly
Significant Difference (HSD) test to compare the axonal morphologies among the five
neuronal subtypes. The results of these analyses confirmed the distinct morphological
characteristics we initially reported. Given the extensive nature of the comparisons,
with ten unique sets arising from the five subtypes, it was not practical to display all
statistical results directly on the figure. To maintain clarity, we have chosen to
summarize these results in **Table S4**, which was referenced in the figure caption. This
table presented the statistical significance of the differences in axonal morphology
between each pair of neuronal subtypes, ensuring that our statement was fully
supported by quantitative evidence.

*7. Fig 2e,f, The Y-axis in the sub-fig (the total axonal length, max axonal length, and*
*total dendrite length) should be corrected as length(mm).*

**Response #12:** We have correct it in the revised figure.

*8. Page 11, line 14, The author mentioned that “The observed brain states exhibited*
*highly similar but not identical patterns of interhemispheric functional connectivity”*
*in fig 3. This conclusion seems unconvincing, and it doesn't make any sense. It seems*
*there is no significant difference between intrahemispheric and interhemispheric*
*connections. The correlation matrixes are also highly similar between the 3 states. Is*
*it necessary to distinguish the three kinds of brain states? I would suggest performing*
*a shuffled test or controlling experiment to validate that the correlation is not an*
*artifact result of background or crosstalk of pixels.*

**Response #13:** (See also **Response #4** to **Reviewer #2**) The functional connectivity
(FC) patterns across different brain states—awake, REM, and non-REM—were
visually similar in terms of the brain regions involved (**Fig. R4B**). However, our
detailed quantitative analysis uncovered critical differences in the strength of these
connections, which were not immediately apparent from a visual inspection of the FC
matrices.

To elucidate these differences, we performed a pairwise subtraction of the FC
matrices corresponding to each of the three states. Our results, illustrated in **Fig. R4C**
and **Fig. 5C**, revealed significant variations in the strength of numerous connections,
particularly when comparing the awake and REM states to the non-REM state ($p <$
0.05 , **page 15: 327-330**).

*9. Fig 3b, the brain activity patterns are characteristic in each brain state, while the*
*connection map in fig3c is highly similar. It looks like the exact opposite conclusion.*
*Moreover, the author did not mention those two sub-figs in the manuscript text. Please*

*explain it in the main text.*

**Response #14:** Refer to **Response #13** to **Reviewer #2** for a detailed comparison of
the states. The original figure was updated with a revised version (**Fig. 5**). Each
component of this new figure was thoroughly described in the text of the manuscript.

*10. On page 12, line 21, 'Similar results were obtained when comparing FC strengths*
*in individual brain states.' But in Figure 4a, only in the non-REM state, the*
*heterotopic connections are significantly higher than intrahemispheric connections.*

**Response #15:** We revised the statement to "*Similar results were obtained when*
*comparing FC strengths in individual brain states (awake, NREM, and REM) (Fig.*
*S8C-E), although in REM state the difference was not significant.*" (**page17: 370-372**)

*11. I am also concerned about the accuracy of brain atlas registration. The captions*
*for A and B are incorrect as shown in fig.S6. In terms of CCF registration, how are*
*the landmarks identified on the exposed cortex surface? Nine markers (depicted as*
*red dots) were manually annotated on each frame, and how to accurately determine*
*the position of these 9 points. I might have missed it but I couldn't find the information*
*about the size of the cranial window. How to make sure the cortex is completely*
*exposed to determine the outer edge of the cortex surface?*

**Response #16:**

1) For CCF (Common Coordinate Framework) registration, we strategically selected
nine markers based on distinct anatomical landmarks. Four of these markers were the
left olfactory bulb (lOB), right olfactory bulb (rOB), center olfactory bulb (cOB), and
the base of the retrosplenial cortex (RSP). These four points were easily identifiable
due to their distinct anatomical features. The remaining five markers were determined
relative to these primary points: one at the midpoint between the cOB and base RSP
(termed 'center marker'), two at the intersections of the cortex's left and right edges

with a line parallel to the IOB-cOB-rOB axis passing through the base RSP, and the
final two at similar intersections but along a line through the center marker. This
registration aimed to align the images with the CCF by outlining the cortex. **Fig. S6B**
include legends for these nine markers for clarity.

2) The cranial window size was carefully designed to ensure complete dorsal cortex
imaging. It extended from 2 mm anterior to the line connecting both eyes to the line
connecting both ears in the anterior-posterior direction, and laterally to the margins of
the ears, thus covering the entire dorsal cortex from the cerebellum to the olfactory
bulbs. **Fig. R7** illustrates the cranial window's range, emphasizing its adequacy in
exposing the entire dorsal cortex.

**Figure R7. Illustration of the range of the cranial window.** Left: four crossing symbols
represent the range in the anterior-posterior direction (from 2 mm in front of the line connecting
two eyes to the line connecting two ears, red dashed lines) and in the left-right direction (from the
margin of the left ear to the margin of the right ear, blue dashed lines). Right: The complete dorsal
cortex exposed within the cranial window. We referred to Couto et.al., 2021 in the surgery of
opening cranial window⁴, and this illustration of cranial window was adopted from Couto et.al.,
2021, Figure 5.

*12. Fig4a, "The left column presents the results of functional connectivity from all*
*anatomically connected regions, which have more intrahemispheric connections than*
*3 heterotopic connections." Is that correct? Again, seems the brain states have no*
*effect on the functional connectivity.*

**Response #17:** Yes, that is correct. We measured FC among anatomically connected

regions as identified in the Allen population data, employing network densities of
69.93% for intrahemispheric-heterotopic and 34.56% for interhemispheric
connections, in line with previously established thresholds³. As our analysis identified
a higher density of anatomical connections in single-neuron data, we also applied an
adjusted network density threshold of 91.22% for intrahemispheric-heterotopic
connections and 70.26% for interhemispheric connections. Both adjusted and
unadjusted threshold analyses yielded similar results (**Fig. 6C-D** and **Fig. S8B-C**).
The original Fig. 4A reported the results from the unadjusted threshold, including
1454 intra-hemispheric connections and 726 heterotopic connections.

The impact of brain states on functional connectivity (FC) was found to be significant.
While the FC patterns of intrahemispheric heterotopic, interhemispheric heterotopic,
and homotopic connections across wakefulness, REM sleep, and non-REM sleep
appeared visually similar, statistical analysis was required to discern nuanced
differences. A t-test was applied to evaluate the variations among these connection
types when transitioning between any two states. This analysis necessitated the
computation of the mean of absolute FC strength differences for each connection type,
thereby ensuring that the averaging process accounted for the magnitude of changes
irrespective of their direction. The statistical results indicated a significant alteration
in FC patterns among the different types of connections when comparing brain states
($p < 0.005$ for all comparisons, **Fig. R8B, Fig. S7B, page 15: 328-329**).

**Figure R8. The FC patterns of intrahemispheric and interhemispheric connections from**
 **wide field data. (A)** FC strength comparison across intrahemispheric-heterotopic (intra-hete),
 interhemispheric-heterotopic (inter-hete), and homotopic (homo) connections. The left column
 presents the results of FC from all anatomically connected regions, which have more
 intrahemispheric-heterotopic connections than interhemispheric-heterotopic connections. The right
 column compares interhemispheric-heterotopic connections with their corresponding
 intrahemispheric-heterotopic counterparts. **(B)** Assessment of the absolute FC strength differences
 between brain states for each type of connection. The left panel illustrates the differences when
 considering all connections, while the right panel isolates these differences for paired connections
 only. Both panels provide a direct comparison between states (wake-REM, wake-nonREM, and
 REM-nonREM).

*13. For computation modeling, why the T_i - T_c overlap is set at 0.4. Does this mean*
 *that different parameter settings will lead to completely different conclusions? The*
 *author built the anatomical patterns model and neuronal network model under*
 *different types of data. It is still not clear to me what the conclusion is. Heterotopic*
 *upstream projections play a pivotal role in regulating the strength of homotopic*
 *functional connectivity. This kind of conclusion is very vague.*

**Response #18:** In response to the concerns regarding our computational modeling

approach, we have revised our manuscript to include a more straightforward and
directly relevant model. Notably, the concept of 'Ti-Tc overlap' from the original
model has been excluded in this updated version.

To address your specific query about the 'Ti-Tc overlap' parameter set at 0.4 in the
original model: this setting was initially chosen to demonstrate the alignment of
neuronal avalanche power-law fitting with homotopic functional connectivity. Upon
further analysis, we adjusted the 'Ti-Tc overlap' parameter and observed that while the
network's criticality remained stable, there was a noticeable increase in functional
connectivity (FC) between homotopic regions as the 'Ti-Tc overlap' value increased.
This observation suggested a non-critical influence of this parameter on our
conclusions.

In both the original and revised manuscripts, the structure of the network model was
influenced by a parameter representing the overlap between projecting neuron groups.
In the original manuscript, this parameter was referred to as 'Ti-Tc overlap' and
'divergence,' whereas in the revised version, we have consistently termed it
'heterogeneity.' We conducted experimental investigations on this heterogeneity
parameter using single-neuron data. Our findings indicated that projections to
different homotopic region pairs exhibited varying degrees of heterogeneity. Notably,
projections to contralateral mutually heterotopic regions demonstrated greater
heterogeneity than those to ipsilateral regions.

In the modeling section of our updated manuscript, we explored the influence of
heterogeneity in upstream projections versus the strength of direct mutual connections
on homotopic FC. Our analysis revealed that the impact of upstream projection
heterogeneity was more significant than that of direct mutual connection strength,
highlighting the crucial role of heterotopic upstream projections in regulating
homotopic FC strength. This modeling conclusion is corroborated by our
experimental analyses, providing a clearer and more precise understanding of the

underlying mechanisms.

*14. In Figure 5, I was wondering in the simulated network, is FC also correlated with*
*T_i-T_c overlap instead of homotopic projection strength?*

**Response #19:** In the revised manuscript, we compared the effect of heterogeneity
(replacing T_i-T_c overlap in the original manuscript) and strength of direct homotopic
connection. The latter was represented by two parameters: homotopic strength
(probability of direct homotopic connections) and self-homotopic overlap (overlap of
self-projection and homotopic connection of an upstream region). The increase in
either homotopic strength or self-homotopic overlap indicates stronger homotopic
projection. We tuned these three parameters respectively in the model and found that
the homotopic correlations were negatively correlated with heterogeneity and
positively correlated with self-homotopic overlap and homotopic strength.
Interestingly, the range of variation in correlation values attributed to heterogeneity
exceeded those due to homotopic strength and self-homotopic overlap (**Fig. R2B-D,**
**Fig. 3C-E, page 12 line 260 - page 13 line 269**). This might be a consequence of the
relative abundance of heterotopic projections compared to homotopic ones;
specifically, in a network with $N=10$ regions, each region received $(N-1)\times 2=18$
heterotopic projections versus a single homotopic projection. To further probe this
phenomenon, we analyzed a minimal network model with just two pairs of homotopic
regions ($N=2$), where the number of homotopic projections mostly approximated that
of heterotopic projections. In this minimal model, the variation range caused by
self-homotopic overlap and homotopic strength was comparable to that caused by
heterogeneity (**Fig. R2B-D, Fig. 3C-E, page 12 line 260 - page 13 line 269**).
Furthermore, the differences between the variation range caused by heterogeneity and
homotopic strength or self-homotopic overlap, denoted as $\Delta(\text{variation range})$, kept
expanding as the number of regions N in the model increased (**Fig. R2E-F, Fig. 3F-G,**
**page 13: 269-274**). This suggested that the impact of heterogeneity became more
pronounced with the increase of projection heterogeneity in the network. Considering

the mouse cortex, which comprised numerous interconnected brain regions (N was
large), we can infer that the heterogeneity of upstream projections was the dominant
factor in shaping homotopic correlation, outweighing the effects of direct connections
between homotopic regions.

*15. On page 17, line1-9, the author proved that lower divergence leads to criticality,*
*but how criticality leads to lower FC variability (more robust connectivity) needs*
*further clarification.*

**Response #20:** We have recognized the confusion caused by the concept of criticality
and our attempts to associate it with FC variability. Therefore, in the updated
manuscript, we have abandoned the concept of criticality and applied more relevant
model indices: the mean value (strength) and standard deviation (variability) of the
Pearson correlations between brain regions' simulated activities. The original model
of self-organized criticality has been replaced with a simpler and more direct network
model. See **Response #4 to Reviewer #1.**

*16. The font in all sub-figures is different (fig 5d-h, fig6). The coordinates of some*
*diagrams are difficult to distinguish.*

**Response #21:** We thank you for the reminder. In the revised manuscript, we have
kept the font the same for all sub-figures and made the coordinates large enough to
recognize.

*17. Page 17, line 6, Please keep the sentences in the same tense.*

**Response #22:** We have paid more attention to the tense of sentences in the revised
manuscript.

**References**

- 1. Shen, K. *et al.* Stable long-range interhemispheric coordination is supported by direct
anatomical projections. *Proc. Natl. Acad. Sci. U.S.A.* **112**, 6473–6478 (2015).
- 2. Hutt, A., Rich, S., Valiante, T. A. & Lefebvre, J. Intrinsic neural diversity quenches the
dynamic volatility of neural networks. *Proc Natl Acad Sci U S A* **120**, e2218841120 (2023).
- 3. Harris, J. A. *et al.* Hierarchical organization of cortical and thalamic connectivity. *Nature* **575**,
195–202 (2019).
- 4. Couto, J. *et al.* Chronic, cortex-wide imaging of specific cell populations during behavior. *Nat*
*Protoc* **16**, 3241–3263 (2021).

Reviewers' Comments:

Reviewer #1:

Remarks to the Author:

The authors have made an excellent revision. They have responded to all my concerns. The manuscript reads now much better and the data is well presented and the conclusions supported. I thank the authors for their revision. I consider the manuscript to be suitable for publication.

Reviewer #2:

Remarks to the Author:

In the present revision, the authors have addressed most of my major concerns. I still have some suggestions for improvement before this paper published.

1. All the figures and corresponding notes in this article need to be carefully rechecked. For example: Figure 5b, where is the color bar for the EEG spectrum? And the frequency scale also needs to be marked. The scale bar of EMG signals, the caption of color bar should be noted in the corresponding figures.

2. For the awake, REM, NREM states, the author mentioned that "our detailed quantitative analysis uncovered critical differences in the strength of these connections, which were not immediately apparent from a visual inspection of the FC matrices", Please quantify the variations in the main manuscript. Also, how about the variation between intrahemispheric and interhemispheric at the stage of REM or NREM. Is there any significant index to evaluate the variation between intra and inter matrices.

3. The font and expression should be consistent. for example: REM, rem....

**Responses to Reviewer #1**

*The authors have made an excellent revision. They have responded to all my concerns.*
*The manuscript reads now much better and the data is well presented and the*
*conclusions supported. I thank the authors for their revision. I consider the*
*manuscript to be suitable for publication.*

**General response:** The reviewer's feedback had significantly improved the quality of
our article, for which we express heartfelt gratitude.

**Responses to Reviewer #2**

*In the present revision, the authors have addressed most of my major concerns.*

*I still have some suggestions for improvement before this paper is published.*

**Response #1:** We greatly appreciate the reviewer's thorough examination of our
manuscript and detailed suggestions for improvement.

*Major Comments*

*1. All the figures and corresponding notes in this article need to be carefully*
*rechecked. For example: Figure 5b, where is the color bar for the EEG spectrum?*
*And the frequency scale also needs to be marked. The scale bar of EMG signals, the*
*caption of color bar should be noted in the corresponding figures.*

**Response #2:** We are very grateful for the reviewer's reminders. In the revised
manuscript, we added the color bar and frequency scale of the EEG spectrum, see

revised Fig. 5B.

**Figure 5. (B) The EEG spectrum for the three brain states.** In the upper row, the rightmost
color bar indicates the amplitude of different frequencies (from 0 Hz to 30 Hz, leftmost). In the
bottom row, we label the scale of 0.2 mV for EMG signals and 50 s for both EEG and EMG.

*2. For the awake, REM, NREM states, the author mentioned that "our detailed*
*quantitative analysis uncovered critical differences in the strength of these*
*connections, which were not immediately apparent from a visual inspection of the FC*
*matrices", Please quantify the variations in the main manuscript. Also, how about the*
*variation between intrahemispheric and interhemispheric at the stage of REM or*
*NREM. Is there any significant index to evaluate the variation between intra and inter*
*matrices.*

**Response #3:** We appreciate the reviewer’s feedback. We added the quantification in
the page 16 lines 331-338 of the revised manuscript: *“Specifically, when comparing*
*with NREM state, a significant increase in FC strength was observed in 91.61% of*
*intrahemispheric connections during the awake state and in 62.80% during the REM*
*state. In terms of interhemispheric connections, 90.22% during awake and 57.96%*
*during REM states showed significantly greater FC strength compared to the NREM*
*state. Furthermore, we identified that 37.42% of intrahemispheric heterotopic*
*connections in the awake state, 42.58% in the NREM state, and 43.12% in the REM*
*state exhibited significantly greater strength than interhemispheric heterotopic*
*connections (Fig. S7A, $p < 0.05$).”* The quantification demonstrated that the *“although*
*exhibiting similar patterns of inter-regional FC (Fig. S7A), different states differed in*
*FC strengths (Fig. 5C and Fig. S7B).”*

*3. The font and expression should be consistent. for example: REM, rem....*

**Response #4:** A thorough review of the manuscript was undertaken to verify the
uniformity of fonts and expressions.
